# GENERALIZED SUM POOLING FOR METRIC LEARNING

## ABSTRACT

A common architectural choice for deep metric learning is a convolutional neural network followed by global average pooling (GAP). Albeit simple, GAP is a highly effective way to aggregate information. One possible explanation for the effectiveness of GAP is considering each feature vector as representing a different semantic entity and GAP as a convex combination of them. Following this perspective, we generalize GAP and propose a learnable generalized sum pooling method (GSP). GSP improves GAP with two distinct abilities: i) the ability to choose a subset of semantic entities, effectively learning to ignore nuisance information, and ii) learning the weights corresponding to the importance of each entity. Formally, we propose an entropy-smoothed optimal transport problem and show that it is a strict generalization of GAP, *i.e.*, a specific realization of the problem gives back GAP. We show that this optimization problem enjoys analytical gradients enabling us to use it as a direct learnable replacement for GAP. We further propose a zero-shot loss to ease the learning of GSP. We show the effectiveness of our method with extensive evaluations on 4 popular metric learning benchmarks. Code is available at: GSP-DML Framework

## 1 INTRODUCTION

Distance metric learning (DML) addresses the problem of finding an embedding function such that the semantically similar samples are embedded close to each other while the dissimilar ones are placed relatively apart in the Euclidean sense. Although the prolific and diverse literature of DML includes various architectural designs (Kim et al., 2018; Lin et al., 2018; Ermolov et al., 2022), loss functions (Musgrave et al., 2020), and data-augmentation techniques (Roth et al., 2020; Venkataramanan et al., 2022), many of these methods have a shared component: a convolutional neural network (CNN) followed by a global pooling layer, mostly global average pooling (GAP) (Musgrave et al., 2020).

Common folklore to explain the effectiveness of GAP is considering each pixel of the CNN feature map as corresponding to a separate semantic entity. For example, spatial extent of one pixel can correspond to a *"tire"* object making the resulting feature a representation for *"tireness"* of the image. If this explanation is correct, the representation space defined via output of GAP is a convex combination of semantically independent representations defined by each pixel in the feature map. Although this folklore is later empirically studied in (Zeiler & Fergus, 2014; Zhou et al., 2016; 2018, and references therein) and further verified for classification in (Xu et al., 2020), its algorithmic implications are not clear. If each feature is truly representing a different semantic entity, should we really average over all of them? Surely, some classes belong to the background and should be discarded as nuisance variables. Moreover, is uniform average of them the best choice? Aren't some classes more important than others? In this paper, we try to answer these questions within the context of metric learning. We propose a learnable and generalized version of GAP which learns to choose the subset of the semantic entities to utilize as well as weights to assign them while averaging.

In order to generalize the GAP operator to be learnable, we re-define it as a solution of an optimization problem. We let the solution space to include 0-weight effectively enabling us to choose subset of the features as well as carefully regularize it to discourage degenerate solution of using all the features. Crucially, we rigorously show that the original GAP is a specific case of our proposed optimization problem for a certain realization. Our proposed optimization problem closely follows optimal transport based *top-k* operators (Cuturi et al., 2019) and we utilize its literature to solve it. Moreover, we present an algorithm for an efficient computation of the gradients over this optimization problem enabling learning. A critical desiderata of such an operator is choosing subset of features

which are discrimantive and ignoring the background classes corresponding to nuisance variables. Although supervised metric learning losses provide guidance for seen classes, they carry no such information to generalize the behavior to unseen classes. To enable such a behavior, we adopt a *zero-shot prediction loss* as a regularization term which is built on expressing the class label embeddings as a convex combination of attribute embeddings (Demirel et al., 2017; Xu et al., 2020).

In order to validate the theoretical claims, we design a synthetic empirical study. The results confirm that our pooling method chooses better subsets and improve generalization ability. Moreover, our method can be applied with any DML loss as GAP is a shared component of them. We applied our method on 6 DML losses and test on 4 datasets. Results show consistent improvements with respect to direct application of GAP as well as other pooling alternatives.

## 2 RELATED WORK

We discuss the works which are most related to ours. Briefly, our contributions include that $i)$ we introduce a general formulation for weighted sum pooling, $ii)$ we formulate local feature selection as an optimization problem which admits closed form gradient expression without matrix inversion, and $iii)$ we propose a meta-learning based zero-shot regularization term to explicitly impose unseen class generalization to the DML problem.

**DML.** Primary thrusts in DML include $i)$ tailoring pairwise loss terms (Musgrave et al., 2020) that penalize the violations of the desired intra- and inter-class proximity constraints, $ii)$ pair mining (Roth et al., 2020), $iii)$ generating informative samples (Ko & Gu, 2020; Liu et al., 2021; Gu et al., 2021; Venkataramanan et al., 2022), and $iv)$ augmenting the mini-batches with virtual embeddings called *proxies* (Wang et al., 2020; Teh et al., 2020). To improve generalization; learning theoretic ideas (Dong et al., 2020; Lei et al., 2021; Gurbuz et al., 2022), disentangling class-discriminative and class-shared features (Lin et al., 2018; Roth et al., 2019), intra-batch feature aggregation (Seidenschwarz et al., 2021), and further regularization terms (Jacob et al., 2019; Zhang et al., 2020; Kim & Park, 2021; Roth et al., 2022) are utilized. To go beyond of a single model, ensemble (Xuan et al., 2018; Kim et al., 2018; Sanakoyeu et al., 2019; Zheng et al., 2021a;b) and multi-task based approaches (Milbich et al., 2020; Roth et al., 2021) are also used. Different to them, we propose a learnable pooling method for the global feature extraction generalizing GAP, a shared component of all of the mentioned works. Hence, our work is orthogonal to all of these and can be used jointly with any of them.

**Prototype-based pooling.** Most related to ours are trainable VLAD (Arandjelovic et al., 2016) and optimal transport based aggregation (Mialon et al., 2021). Such methods employ similarities to the prototypes to form a vector of aggregated local features for each prototype and build ensemble of representations. Similar to us, Mialon et al. (2021) uses optimal transport formulation to select local features to be pooled for each prototype. That said, such methods map a set of features to another set of features without discarding any and do not provide a natural way to aggregate the class-discriminative subset of the features. On the contrary, our pooling machine effectively enables learning to select discriminative features and maps a set of features to a single feature that is distilled from nuisance information.

**Attention-based pooling.** Among the methods that reweights the CNN features before pooling, CroW (Kalantidis et al., 2016), Trainable-SMK (Tolias et al., 2020), and CBAM (Woo et al., 2018) build on feature magnitude based saliency, assuming that the backbone functions must be able to zero-out nuisance information. Yet, such a requirement is restrictive for the parameter space and annihilation of the non-discriminative information might not be feasible in some problems. Similarly, attention-based weighting methods DeLF (Noh et al., 2017), GSoP (Gao et al., 2019) do not have explicit control on feature selection behavior and might result in poor models when jointly trained with the feature extractor (Noh et al., 2017). Differently, our method unifies attention-based feature masking practices (*e.g. convolution*, *correlation*) with an efficient-to-solve optimization framework and lets us do away with engineered heuristics in obtaining the masking weights (*e.g. normalization*, *sigmoid*, *soft-plus*) without restricting the solution space unlike magnitude based methods.

**Optimal transport based operators.** Optimal transport (OT) distance (Cuturi, 2013) to match local features is used as the DML distance metric instead of $\ell2$ in (Zhao et al., 2021). Despite effective, replacing $\ell2$ with OT increases memory cost for image representation as well as computation cost for

the distance computation. Different to them, we shift OT based computation in pooling (*i.e.*, feature extraction) stage while having OT's merits and hence, do not affect the memory and computation costs of the inference by sticking to $\ell 2$ metric. Moreover, our feature selection and aggregation formulation has close relation to optimal transport (Cuturi, 2013) based top-$k$ (Xie et al., 2020), ranking (Cuturi et al., 2019) and aggregation (Mialon et al., 2021) operators. What makes our method different is the unique way we formulate the feature selection problem to fuse aggregation into it. Our formulation allows computationally appealing and matrix inversion free gradient computation of the selection operator unlike optimal transport plan based counterparts (Luise et al., 2018).

## 3 PRELIMINARIES

Consider the data distribution $p_{\mathcal{X}\times\mathcal{Y}}$ over $\mathcal{X}\times\mathcal{Y}$ where $\mathcal{X}$ is the space of data points and $\mathcal{Y}$ is the space of labels. Given *iid.* samples from $p_{\mathcal{X}\times\mathcal{Y}}$ as $\{(x_i, y_i)\}$, distance metric learning problem aims to find the parameters $\theta$ of an embedding function $e(\cdot; \theta) : \mathcal{X} \to I\!\!R^d$ such that the Euclidean distance in the space of embeddings is consistent with the label information where $d$ is the embedding dimension. More specifically, $\|e(x_i; \theta) - e(x_j; \theta)\|_2$ is small whenever $y_i = y_j$, and large whenever $y_i \neq y_j$. In order to enable learning, this requirement is represented via loss function $l((x_i, y_i), (x_j, y_j); \theta)$ (*e.g.* *contrastive* (Wu et al., 2017), *triplet* (Schroff et al., 2015), *multi-similarity* (Wang et al., 2019)).

The typical learning mechanism is gradient descent of an empirical risk function defined over a batch of data points $B$. To simplify notation throughout the paper, we will use $b = \{b(i) \mid x_i, y_i \in B\}_i$ to index the samples in a batch. Then, the typical empirical risk function is defined as:

$$\mathcal{L}_{DML}(b; \theta) := \frac{1}{|b|^2} \sum_{i \in b} \sum_{j \in b} l((x_i, y_i), (x_j, y_j); \theta). \tag{3.1}$$

We are interested specific class of embedding functions where a global average pooling is used. Specifically, consider the composite function family $e = g \circ f$ such that $g$ is pooling and $f$ is feature computation. We assume a further structure over the functions $g$ and $f$. The feature function $f$ maps the input space $\mathcal{X}$ into $I\!\!R^{w \times h \times d}$ where $w$ and $h$ are spatial dimensions. Moreover, $g$ performs averaging as;

$$g(f(x; \theta)) = \frac{1}{w \cdot h} \sum_{i \in [w \cdot h]} f_i, \tag{3.2}$$

where $[n] = 1, \ldots, n$ and we let $f_i \in I\!\!R^d$ denote $i^{th}$ spatial feature of $f(x; \theta)$ to avoid convoluted notation. In the rest of the paper, we generalize the pooling function $g$ into a learnable form and propose an algorithm to learn it.

## 4 METHOD

Consider the pooling operation in Eq. (3.2), it is a simple averaging over pixel-level feature maps ($f_i$). As we discuss in § 1, one explanation for the effectiveness of this operation is considering each $f_i$ as corresponding to a different semantic entity corresponding to the spatial extend of the pixel, and the averaging as convex combination over these semantic classes. Our method is based on generalizing this averaging such that a specific subset of pixels (correspondingly subset of semantic entities) are selected and their weights are adjusted according to their importance.

We generalize Eq. (3.2) in § 4.1 by formulating a feature selection problem in which we prioritize a subset of the features that are closest to some trainable prototypes. If a feature is to be selected, its weight will be high. We then formulate our pooling operation as a differentiable layer so that the prototypes can be learned along with the rest of the embedding function parameters in § 4.2. We learn the prototypes with class-level supervision, however in metric learning, learned representations should generalize to unseen classes. Thus, we introduce a zero-shot prediction loss to regularize prototype training for zero-shot setting in § 4.3.

## 4.1 Generalized Sum Pooling as a Linear Program

Consider the pooling function, $g$, with adjustable weights as $g(f(x;\theta);\omega) = \sum_{i\in[n]} p_i f_i$ where $n = w\,h$. Note that, $p_i = 1/n$ corresponds to average pooling. Informally, we want to control the weights to ease the metric learning problem. Specifically, we want the weights corresponding to background classes to be 0 and the ones corresponding to discriminative features to be high.

If we were given representations of discrimantive semantic entities, we could simply compare them with the features ($f_i$) and choose the ones with high similarity. Our proposed method is simply learning these representations and using them for weight computations. We first discuss the weight computation part before discussing learning the representations of prototypes.

Assume that there are $m$ discrimantive semantic entities which we call *prototypes* with latent representations $\omega = \{\omega_i\}_{i\in[m]}$ of appropriate dimensions (same as $f_i$). Since we know that not all features ($\{f_i\}_{i\in[n]}$) are relevant, we need to choose a subset of $\{f_i\}_{i\in[n]}$. We perform this top-$k$ selection process by converting it into an optimal transport (OT) problem.

Consider a cost map $c_{ij} = \|\bar\omega_i - \bar f_j\|_2$ which is an $m$ (number of prototypes) by $n$ (number of features) matrix representing the closeness of prototypes $\omega_i$ and features $f_j$ after some normalization $\bar u = u/\max\{1,\|u\|_2\}$. We would like to find a transport map $\pi$ which re-distributes the uniform mass from features to prototypes. Since we do not have any prior information over features, we also consider its marginal distribution (importance of each feature to begin with) to be uniform. As we need to choose a subset, we set $\mu\in[0,1]$ ratio of mass to be transported. The resulting OT problem is:

$$\rho^*, \pi^* = \underset{\substack{\rho,\pi\geqslant 0 \\ \rho_j + \Sigma_i \pi_{ij} = 1/n \\ \Sigma_{ij}\pi_{ij} = \mu}}{\arg\min} \sum_{ij} c_{ij}\pi_{ij}. \tag{P1}$$

Different to typical OT literature, we introduce decision variables, $\rho$, to represent residual weights to be discarded. Specifically modelling discarded weight instead of enforcing another marginalization constraint is beneficial beyond stylistic choices as it allows us to very efficient compute gradients. When the introduced transport problem is solved, we perform weighting using residual weights as:

$$g(f(x;\theta);\omega) = \sum_i p_i f_i = \sum_i \frac{1/n - \rho_i^*}{\mu} f_i \tag{4.1}$$

Given set of prototypes $\{\omega_i\}_{i\in[m]}$, solving the problem in (P1) is a strict generalization of GAP since setting $\mu = 1$ recovers the original GAP. We formalize this equivalence in the following claim.

**Claim 4.1.** *If $\mu = 1$, the operation in Eq.* (4.1) *reduces to global average pooling in Eq.* (3.2).

We defer the proof to Appendix. Having generalized GAP to a learnable form, we introduce a method to learn the prototypes $\{\omega_i\}_{i\in[m]}$ in the next section.

## 4.2 Generalized Sum Pooling as a Differentiable Layer

Consider the generalized form of pooling, defined as solution of (P1), as a layer of a neural network. The input is the feature vectors $\{f_i\}_{i\in[n]}$, the learnable parameters are prototype representations $\{\omega_i\}_{i\in[m]}$, and the output is residual weights $\rho^*$. To enable learning, we need partial derivatives of $\rho^*$ with respect to $\{\omega_i\}_{i\in[m]}$. However, this function is not smooth. More importantly it requires the $\mu$ parameter to be known a priori.

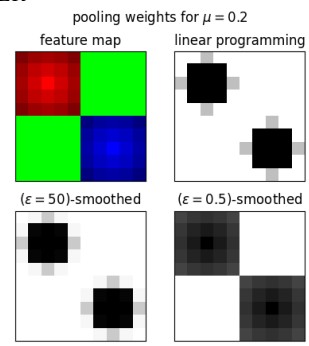

We use a toy example to set the stage for rest of the formulation. Consider a 10x10x3 feature map visualized as RGB-image in Fig. 1 and corresponding two prototypes with representations $(1,0,0)$ (red) and $(0,0,1)$ (blue). The true $\mu = 0.5$ since the half of the image corresponds to red and blue, and other half is background class of green. Consider an under-estimation of $\mu = 0.2$, the global solution (shown as linear programming) is explicitly ignoring informative pixels (part of red and blue region). To solve this issue,

Figure 1: Feature map and the resultant pooling weights (higher the darker) of different problems.

we use entropy smoothing which is first introduced in (Cuturi, 2013) to enable fast computation of optimal transport. Consider the entropy smoothed version of the original problem in (P1) as:

$$\rho^{(\varepsilon)}, \pi^{(\varepsilon)} = \underset{\substack{\rho, \pi \geqslant 0 \\ \rho_j + \Sigma_i \pi_{ij} = 1/n \\ \Sigma_{ij} \pi_{ij} = \mu}}{\arg \min} \sum_{ij} c_{ij} \pi_{ij} + \frac{1}{\varepsilon} \left( \sum_{ij} \pi_{ij} \log \pi_{ij} + \sum_j \rho_j \log \rho_j \right), \tag{P2}$$

and obtain pooling weights by replacing $\rho^*$ with $\rho^{(\varepsilon)}$ in Eq. (4.1). When smoothing is high ($\varepsilon \to 0$), the resulting solution is uniform over features similar to GAP. When it is low, the result is similar to top-$k$ like behavior. For us, $\varepsilon$ controls the trade-off between picking $\mu$ portion of the features that are closest to the prototypes and including as much features as possible for weight transfer.

We further visualize the solution of the entropy smoothed problem in Fig. 1 showing desirable behavior even with underestimated $\mu$.

Beyond alleviating the under-estimation of $\mu$ problem, entropy smoothing also makes the problem strictly convex and smooth. Thus, the solution of the problem enables differentiation and in fact, admits closed-form gradient expression. We state the solution of (P2) and their corresponding gradients in the following propositions and defer their proofs to Appendix.

**Proposition 4.1.** *Given initialization $t^{(0)} = 1$, consider the following iteration:*

$$\rho^{(k+1)} = 1/n \left( 1 + t^{(k)} \exp(\text{-}\varepsilon c)^{\mathsf{T}} \mathbf{1}_m \right)^{-1}, \ \ t^{(k+1)} = \mu \left( \mathbf{1}_m^{\mathsf{T}} \exp(\text{-}\varepsilon c) \rho^{(k+1)} \right)^{-1}$$

*where $\exp$ and $(\cdot)^{-1}$ are element-wise and $\mathbf{1}_m$ is $m$-dimensional vector of ones. Then, $(\rho^{(k)}, t^{(k)})$ converges to the solution of* (P2) *defining transport map via $\pi^{(k)} = t^{(k)} \exp(\text{-}\varepsilon c) Diag(\rho^{(k)})$.*

**Proposition 4.2.** *Consider any differentiable loss function $\mathcal{L}$ as a function of $(\rho, \pi)$. Given gradients $\frac{\partial \mathcal{L}}{\partial \rho^{(\varepsilon)}}$ and $\frac{\partial \mathcal{L}}{\partial \pi^{(\varepsilon)}}$, with $q = \rho^{(\varepsilon)} \odot \frac{\partial \mathcal{L}}{\partial \rho^{(\varepsilon)}} + (\pi^{(\varepsilon)} \odot \frac{\partial \mathcal{L}}{\partial \pi^{(\varepsilon)}})^{\mathsf{T}} \mathbf{1}_m$ and $\eta = (\rho^{(\varepsilon)} \odot \frac{\partial \mathcal{L}}{\partial \rho^{(\varepsilon)}})^{\mathsf{T}} \mathbf{1}_n - n \, q^{\mathsf{T}} \rho^{(\varepsilon)}$, the gradient with respect to $c$ reads:*

$$\frac{\partial \mathcal{L}}{\partial c} = \text{-}\varepsilon \left( \pi^{(\varepsilon)} \odot \frac{\partial \mathcal{L}}{\partial \pi^{(\varepsilon)}} - n \pi^{(\varepsilon)} Diag \left( q - \frac{\eta}{1 - \mu - n \rho^{(\varepsilon)\mathsf{T}} \rho^{(\varepsilon)}} \right) \rho^{(\varepsilon)} \right) \quad , \tag{4.2}$$

*where $\odot$ denotes element-wise multiplication.*

Proposition 4.1 and 4.2 suggest that our feature selective pooling can be implemented as a differentiable layer. Moreover, Proposition 4.2 gives a matrix inversion free computation of the gradient with respect to the costs unlike optimal transport based operators (Luise et al., 2018). Thus, the prototypes, $\omega$, can be jointly learned with the feature extraction efficiently.

## 4.3 Cross-batch Zero-shot Regularization

Until now, we formulate a prototype based feature pooling and can learn the prototypes using class labels with any DML loss. To this end, specializing to classes is a feasible behavior for the prototypes. On the other hand, we rather want the prototypes to capture transferable attributes so that the learning can be transferred to the unseen classes as long as the attributes are shared. In other words, learning with prototype based pooling shapes the embedding geometry in such a way that we have clusters corresponding to the prototypes in the embedding space. We want such clusters to have transferable semantics rather than class-specific information. To enable this, we now formulate a mechanism to predict class embedding vectors from the prototype assignment vectors and use that mechanism to tailor a loss regularizing the prototypes to have transferable representations.

Our feature selection layer should learn discriminative feature prototypes, $\omega$, using top-down label information. Consider two randomly selected batches, $(b_1, b_2)$, of data sampled from the distribution. If the prototypes are corresponding to discrimantive entities, the weights transferred to them (*i.e.*, marginal distribution of prototypes) should be useful in predicting the classes and such behavior should be consistent between batches for zero-shot prediction. Formally, if one class in $b_2$ does not exist in $b_1$, a predictor on class labels based on marginal distribution of prototypes for each class of $b_1$ should still be useful for $b_2$. Sadly, DML losses do not carry such information. We thus formulate a zero-shot prediction loss to enforce such zero-shot transfer.

We consider that we are given a semantic embedding vector for each of $c$-many class labels, $\Upsilon = [v_i]_{i \in [c]}$. We are to predict such embeddings from the marginal distribution of the prototypes. In

particular, we use linear predictor, $A$, to predict label embeddings as $\hat{v} = A\,z$ where $z$ is the normalized distribution of the weighs on the prototypes;

$$z = \tfrac{1}{\mu} \sum_i \pi_i^{(\varepsilon)} \quad \text{where} \quad \pi^{(\varepsilon)} = [\pi_i^{(\varepsilon)}]_{i \in [n]}. \tag{4.3}$$

If we consider the prototypes as semantic vectors of some auxiliary labels such as *attributes* commonly used in zero-shot learning (ZSL) (Demirel et al., 2017; Xu et al., 2020; Huynh & Elhamifar, 2020), then we can interpret $z$ as *pseudo-attribute* predictions. Given pseudo-attribute predictions, $\{z_i\}_{i \in b}$, and corresponding class embeddings for a batch, $b$, we fit the predictor as;

$$A_b = \underset{A = [a_i]_{i \in [m]}}{\arg\min} \sum_{i \in b} \|A\,z_i - v_{y_i}\|_2^2 + \epsilon \sum_{i \in [m]} \|a_i\|_2^2. \tag{P3}$$

which admits a closed form expression enabling back propagation $A_b = \Upsilon_b \left(Z_b^\mathsf{T} Z_b + \epsilon I\right)^{-1} Z_b^\mathsf{T}$ where $\Upsilon_b = [v_{y_i}]_{i \in b}$, $Z_b = [z_i]_{i \in b}$. In practice, we are not provided with the label embeddings, $\Upsilon = [v_i]_{i \in [c]}$. Nevertheless, having a closed-form expression for $A_b$ enables us to exploit a meta-learning scheme like (Bertinetto et al., 2018) to formulate a zero-shot prediction loss to learn them jointly with the rest of the parameters.

Specifically, we split a batch, $b$, into two as $b_1$ and $b_2$ such that classes are disjoint. We then estimate attribute embeddings, $A_{b_k}$, according to (P3) using one set and use that estimate to predict the label embeddings of the other set to form zero-shot prediction loss. Formally, our loss becomes:

$$\mathcal{L}_{ZS}(b;\theta) = \tfrac{1}{|b_2|} \sum_{i \in b_2} \log\left(1 + \sum_{j \in [c]} e^{(v_j - v_{y_i})^\mathsf{T} A_1 z_i}\right) + \tfrac{1}{|b_1|} \sum_{i \in b_1} \log\left(1 + \sum_{j \in [c]} e^{(v_j - v_{y_i})^\mathsf{T} A_2 z_i}\right) \tag{4.4}$$

*i.e.*, rearranged *soft-max cross-entropy* where $A_k = A_{b_k}$ with the abuse of notation, and $\theta = \{\theta_f, \omega, \Upsilon\}$ (*i.e.*, CNN parameters, prototype vectors, label embeddings).

We learn attribute embeddings (*i.e.*, columns of $A$) as sub-task and can define such learning as a differentiable operation. Thus, our cross-batch zero-shot prediction loss, $\mathcal{L}_{ZS}$, is to achieve *learning to learn attribute embeddings for zero-shot prediction*. Intuitively, such a regularization should be useful in better generalization of our pooling operation to unseen classes since pseudo-attribute predictions are connected to prototypes and the local features. We combine this loss with the metric learning loss using $\lambda$ mixing (*i.e.*, $(1-\lambda)\mathcal{L}_{DML} + \lambda\mathcal{L}_{ZS}$) and jointly optimize.

### 4.4 IMPLEMENTATION DETAILS

**Embedding function.** For the embedding function, $f(\cdot;\theta)$, we use ResNet20 (He et al., 2016) for Cifar (Krizhevsky & Hinton, 2009) experiments, and ImageNet (Russakovsky et al., 2015) pretrained BN-Inception (Ioffe & Szegedy, 2015) for the rest. We exploit architectures until the output before the global average pooling layer. We add a per-pixel linear transform (*i.e.*, 1x1 convolution), to the output to obtain the local embedding vectors of size 128.

**Pooling layer.** For baseline methods, we use global average pooling. For our method, we perform parameter search and set the hyperparameters accordingly. Specifically, we use 64- or 128-many prototypes depending on the dataset. We use $\varepsilon = 0.5$ for proxy-based losses and $\varepsilon = 5.0$ for non-proxy losses. For the rest, we set $\mu = 0.3$, $\epsilon = 0.05$, $\lambda = 0.1$ and we iterate until $k = 100$ in Proposition 4.1. The embedding vectors upon global pooling are $\ell 2$ normalized to have unit norm.

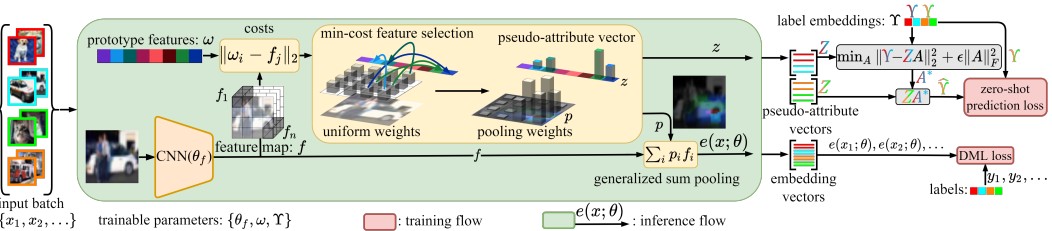

Figure 2: Sketch of the method.

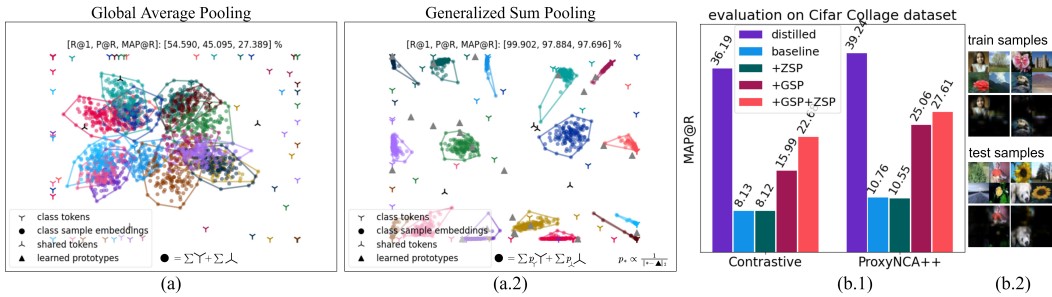

Figure 3: (a): GAP vs GSP in aggregating features, where tokens denote learned embedding vectors and samples are obtained by aggregating them. (b): Evaluation on Cifar Collage dataset (b.1), and (b.2) sample train and test images with their attention maps in terms of pooling weights. *Distilled* denotes baseline performance on non-collage dataset (*i.e.*, excluding the shared classes).

## 5 EXPERIMENTS

We start our empirical study with a synthetic study validating the role of GAP in learning and the impact of GSP on the feature geometry. We further examine the effectiveness of our generalized sum pooling in metric learning for various models and datasets. We further perform ablation studies for the implications of our formulation as well as effects of the hyperparameters. We share the implementation details as well as complete Tensorflow (Abadi et al., 2016) code base in the supplemental materials.

### 5.1 SYNTHETIC STUDY

We design a synthetic empirical study to evaluate GSP in a fully controlled manner. We consider 16-class problem such that classes are defined over trainable tokens. In this setting, tokens correspond to semantic entities but we choose to give a specific working to emphasize that they are trained as part of the learning. Each class is defined with 4 distinct tokens and there are also 4 background tokens shared by all classes. For example, a *"car"* class would have tokens like *"tire"* and *"window"* as well as background tokens of *"tree"* and *"road"*. We sample class representations from both class specific and background tokens according to a mixing ratio $\tilde{\mu} \sim \mathcal{N}(0.5, 0.1)$. Such a 50-many feature collection will correspond to a training sample (*i.e.*, we are mimicking CNN's output with trainable tokens). We then obtain global representations using GAP and GSP. We visualize the geometry of the embedding space in Fig. 3-(a). With GAP, we observe overlapping class convex hulls hence classes are not well discriminated. In other other hand, GSP gives well separated class convex hulls further validation that it learns to ignore background tokens.

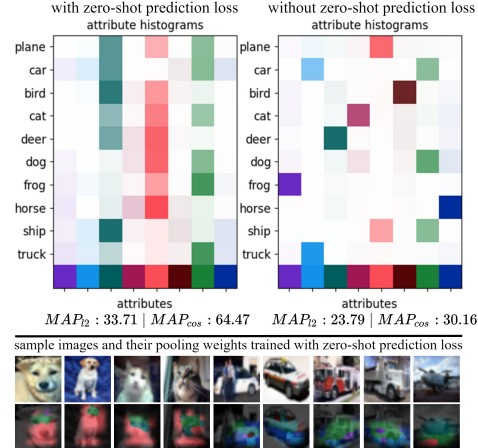

Figure 4: Comparing the distributions of the learned 8 prototypes across classes of Cifar10 dataset with and without $\mathcal{L}_{ZS}$. Pooling weights are coloured according to the dominant prototype at that location.

We further extend this study to image domain. We consider the 20 *super-classes* of Cifar100 dataset where each has 5 sub-classes. For each super-class, we split the sub-classes for train (2), validation (1), and test (2). We consider 4 super-classes as the shared classes and compose $4 \times 4$-stitched collage images for the rest 16 classes. In particular, we sample an image from a class and then sample 3 images from shared classes. We use ResNet20 backbone pretrained on Cifar100 classification task and follow the implementation explained in § 4.4. We provide the evaluation results in Fig. 3-(b). GSP and the proposed zero shot loss effectively increase MAP@R. We also provide sample train and test images to showcase that our pooling can transfer well to unseen domain.

We also evaluate the zero-shot prediction performance of the pseudo-attribute vectors. We train on Cifar10 dataset with 8 prototypes using ProxyNCA++ Teh et al. (2020) (PNCA) loss with and without $\mathcal{L}_{ZS}$. We then extract pseudo-attribute histograms for each class and visualize them in Fig. 4. We observe transferable representations with $\mathcal{L}_{ZS}$ and we visually show in Fig. 4 that the semantic entities represented by the prototypes transfer across classes. We quantitatively evaluate such behavior by randomly splitting the classes into half and apply cross-batch zero-shot prediction explained in § 4.3. Namely, we fit $A$ in (P3) for one subset and use it to predict the class embeddings for the other set. We pre-compute class embeddings from the dataset as the class mean. To this end, our evaluation assesses generalization of both the features and the prototypes. We use MAP with both $\ell 2$ distance and *cosine* similarity in our evaluation. We repeat the experiment 1000 times. We observe in Fig. 4 that zero-shot performance of the prototypes learned with $\mathcal{L}_{ZS}$ is substantially superior. We also see that our feature aggregation method enables approximate localization of the semantic entities. Recent ZSL approaches (Huynh & Elhamifar, 2020; Xu et al., 2020) can provide attribute localization and share a similar spirit with our method. However, attribute annotations must be provided for those methods whereas we exploit only class labels to extract attribute-like features. Our method can be considered as attribute-unsupervised version of these methods.

## 5.2 DEEP METRIC LEARNING EXPERIMENTS

### 5.2.1 SETUP

In order to minimize the confounding of factors other than our proposed method, we keep the comparisons as fair as possible following the suggestions of recent work explicitly studying the fair evaluation strategies for metric learning (Roth et al., 2020; Musgrave et al., 2020; Fehervari et al., 2019). Specifically, we mostly follow the procedures proposed in (Musgrave et al., 2020) to provide fair and unbiased evaluation of our method as well as comparisons with the other methods. We additionally follow the relatively old-fashioned conventional procedure (Oh Song et al., 2016) for the evaluation of our method and provide those results in the supplementary material. We provide full detail of our experimental setup in the supplementary material for complete transparency and reproducibility.

**Datasets.** We use CUB-200-2011 (CUB) (Wah et al., 2011), Cars196 (Krause & Golovin, 2014), In-shop (Liu et al., 2016), and Stanford Online Products (SOP) (Oh Song et al., 2016) with the data augmentation from (Musgrave et al., 2020).

**Evaluation metrics.** We report mean average precision (MAP@R) at R where R is defined for each query and is the total number of true references of the query.

**Hyperparameters.** We use Adam (Kingma & Ba, 2014) optimizer with learning rate $10^{-5}$, weight decay $10^{-4}$, batch size 32 (4 per class). We train 4-fold: 4 models (1 for each $^3/_4$ train set partition).

**Evaluation.** Average performance (128D) where each of 4-fold model is trained 3 times resulting in realization of $3^4{=}81$ different model collections. In our results we provide mean of 81 evaluations.

**Baselines.** We implement our method on top of and compare with *Contrastive (C2)*: Contrastive with positive margin (Wu et al., 2017), *MS*: Multi-similarity (Wang et al., 2019), *Triplet*: Triplet (Schroff et al., 2015), *XBM*: Cross-batch memory (Wang et al., 2020) with contrastive loss (Hadsell et al., 2006), *PNCA*: ProxyNCA++ (Teh et al., 2020), *PAnchor*: ProxyAnchor (Kim et al., 2020).

### 5.2.2 RESULTS

We compare our method (GSP) against direct application of GAP with 6 DML methods in 4 datasets. We also evaluate 13 additional pooling alternatives on *Ciffar Collage* and CUB datasets. We provide the results in supplementary material, Tab. 4. Based on CUB performances, we pick generalized mean pooling (GeMean) (Radenović et al., 2018) and DeLF (Noh et al., 2017) to compare against in 4 DML benchmarks. We also evaluate max pooling (GMP) and its combination with GAP as we typically observe GAP+GMP in the recent works (Venkataramanan et al., 2022; Teh et al., 2020; Kim et al., 2020; Wang et al., 2020). We also apply our method with GMP (GMP+GSP) and with GeMean (GeMean+GSP) to show that per channel selection is orthogonal to our approach and thus, GSP can marginally improve those methods as well.

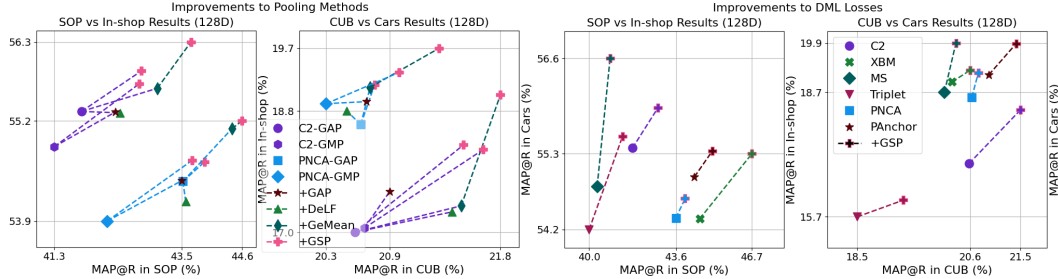

Figure 5: Summary of relative improvements recorded in Table 2.

We provide the detailed evaluation results in supplementary material, Tab. 2 and we summarize the relative MAP@R orderings of the methods with 128D embeddings in Fig. 5. We observe consistent improvements upon direct application of GAP in all datasets. On the average, we consistently improve the baselines $\approx 1\%$ points in MAP@R. Our improvement margins are superior to ones of attention based DeLF pooling. We improve state-of-the-art (SOTA) XBM method up to $2\%$ points, which is a good evidence that application of GSP is not limited to loss terms but can be combined with different DML approaches. We also consistently improve GMP and GeMean pooling methods in all datasets, yet another evidence that our method can be combined with max pooling based methods.

We additionally evaluate our method with different architectures and methods in conventional setting (Oh Song et al., 2016) for the comparison with SOTA. The results are provided in supplementary material, Tab. 3, where we observe that we achieve SOTA performances with XBM (Wang et al., 2020) and LIBC (Seidenschwarz et al., 2021) methods.

### 5.2.3 ABLATIONS

**Effect of $\mathcal{L}_{ZS}$.** We empirically show the effect of $\mathcal{L}_{ZS}$ on learned representations in § 5.1. We further examine the effect of $\mathcal{L}_{ZS}$ quantitatively by enabling/disabling it in 4 datasets. We also evaluate its effect without GSP by setting $\mu=1$ where we use GAP with pseudo-attribute vectors. The results are summarized in Tab. 1 showing that both components improves the baseline and their combination brings the best improvement. We observe similar behavior in *Cifar Collage* experiment (Fig. 3-(b)) where the effect of $\mathcal{L}_{ZS}$ is more substantial.

Table 1: Effects of the two components: Zero-shot prediction loss (ZSP) and Generalized Sum Pooling (GSP)

| Base Method: C2 | | MAP@R | | | | | | | |
|---|---|---|---|---|---|---|---|---|---|
| Component | | SOP | | In-shop | | CUB | | Cars196 | |
| ZSP | GSP | 512D | 128D | 512D | 128D | 512D | 128D | 512D | 128D |
| | | 45.85 | 41.79 | 59.07 | 55.38 | 25.95 | 20.58 | 24.38 | 17.02 |
| | ✓ | 46.78 | 42.66 | 59.46 | 55.50 | 26.25 | 20.85 | 25.54 | 17.88 |
| ✓ | | 46.60 | 42.55 | 59.38 | 55.43 | 26.49 | 21.08 | 25.54 | 17.67 |
| ✓ | ✓ | **46.81** | **42.84** | **60.01** | **55.94** | **27.12** | **21.52** | **26.25** | **18.31** |

**Effect of $\mu$.** As we discuss in § 4.2, GSP is similar to top-$k$ operator with an adaptive $k$ thanks to entropy smoothing. We empirically validate such behavior in CUB dataset with C2 loss by sweeping $\mu$ parameter controlling top-$k$ behavior. We plot the performances in Fig. 6. Relatively lower values of $\mu$ performs similarly. As we increase $\mu$, the performance drops towards GAP due to possibly overestimating the foreground ratio.

## 6 CONCLUSION

Building on perspective explaining the success of GAP, we proposed a learnable and generalized version. Our proposed generalization is a trainable pooling layer that selects the feature subset and re-weight it during pooling. To enable effective learning of the proposed layer, we also proposed a regularization loss to improve zero-shot transfer. With extensive empirical studies, we validated the effectiveness of the proposed pooling layer in various metric learning benchmarks.

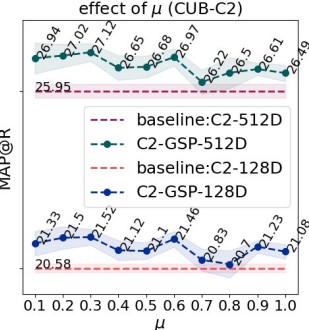

Figure 6: Effect of $\mu$ in CUB dataset with C2 loss. Shaded regions represent $\mp std$.

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

SUPPLEMENTAL MATERIAL FOR "GENERALIZED SUM POOLING FOR METRIC LEARNING"

# 1 EXTENDED EMPIRICAL STUDY FOR DEEP METRIC LEARNING

In the following sections, we explain our empirical study in detail and provide additional experiments on effect of hyperparameters as well as evaluation with the *conventional* experimental settings.

REPRODUCIBILITY

We provide full detail of our experimental setup and recapitulate the implementation details for the sake of complete transparency and reproducibility. Code is available at: GSP-DML Framework.

## 1.1 SETUP

### 1.1.1 DATASETS

We perform our experiments on 4 widely-used benchmark datasets: Stanford Online Products (SOP) (Oh Song et al., 2016), In-shop (Liu et al., 2016), Cars196 (Krause & Golovin, 2014) and, CUB-200-2011 (CUB) (Wah et al., 2011).

**SOP** has 22,634 classes with 120,053 product images. The first 11,318 classes (59,551 images) are split for training and the other 11,316 (60,502 images) classes are used for testing.

**In-shop** has 7,986 classes with 72,712 images. We use 3,997 classes with 25,882 images as the training set. For the evaluation, we use 14,218 images of 3,985 classes as the query and 12,612 images of 3,985 classes as the gallery set.

**Cars196** contains 196 classes with 16,185 images. The first 98 classes (8,054 images) are used for training and remaining 98 classes (8,131 images) are reserved for testing.

**CUB-200-2011** dataset consists of 200 classes with 11,788 images. The first 100 classes (5,864 images) are split for training, the rest of 100 classes (5,924 images) are used for testing.

**Data augmentation** follows (Musgrave et al., 2020). During training, we resize each image so that its shorter side has length 256, then make a random crop between 40 and 256, and aspect ratio between $3/4$ and $4/3$. We resize the resultant image to 227×227 and apply random horizontal flip with 50% probability. During evaluation, images are resized to 256 and then center cropped to 227×227.

### 1.1.2 TRAINING SPLITS

**Fair evaluation.** We split datasets into disjoint training, validation and test sets according to (Musgrave et al., 2020). In particular, we partition $50\%/50\%$ for training and test, and further split training data to 4 partitions where 4 models are to be trained by exploiting $1/4$ as validation while training on $3/4$.

**Conventional evaluation.** Following relatively *old-fashioned* conventional evaluation (Oh Song et al., 2016), we use the whole train split of the dataset for training and we use the test split for evaluation as well as monitoring the training for early stopping.

**Hyperparameter tuning.** For the additional experiments related to the effect of hyperparameters, we split training set into 5 splits and train a single model on the $4/5$ of the set while using $1/5$ for the validation.

### 1.1.3 EVALUATION METRICS

We consider precision at 1 (P@1) and mean average precision (MAP@R) at R where R is defined for each query[1] and is the total number of true references as the query. Among those, MAP@R performance metric is shown to better reflect the geometry of the embedding space and to be less

---

[1]A query is an image for which similar images are to be retrieved, and the references are the images in the searchable database.

noisy as the evaluation metric (Musgrave et al., 2020). Thus, we use MAP@R to monitor training in our experiments except for conventional evaluation setting where we monitor P@1.

*P@1:* Find the nearest reference to the query. The score for that query is 1 if the reference is of the same class, 0 otherwise. Average over all queries gives P@1 metric.

*P@R:* For a query, $i$, find $R_i$ nearest references to the query and let $r_i$ be the number of true references in those $R_i$-neighbourhood. The score for that query is $P@R_i = r_i/R_i$. Average over all queries gives P@R metric, *i.e.*, $P@R = \frac{1}{n} \sum_{i \in [n]} P@R_i$, where $n$ is the number of queries.

*MAP@R:* For a query, $i$, we define $MAP@R_i := \frac{1}{R_i} \sum_{i \in [R_i]} P(i)$, where $P(i) = P@R_i$ if $i^{th}$ retrieval is correct or 0 otherwise. Average over all queries gives MAP@R metric, *i.e.*, $MAP@R = \frac{1}{n} \sum_{i \in [n]} MAP@R_i$, where $n$ is the number of queries.

### 1.1.4 TRAINING PROCEDURE

**Fair evaluation.** We use Adam (Kingma & Ba, 2014) optimizer with constant $10^{-5}$ learning rate, $10^{-4}$ weight decay, and default moment parameters, $\beta_1 = .9$ and $\beta_2 = .99$. We use batch size of 32 (4 samples per class). We evaluate validation MAP@R for every 100 steps of training in CUB and Cars196, for 1000 steps in SOP and In-shop. We stop training if no improvement is observed for 15 steps in CUB and Cars196, and 10 steps in SOP and In-shop. We recover the parameters with the best validation performance. Following (Musgrave et al., 2020), we train 4 models for each $3/4$ partition of the train set. Each model is trained 3 times. Hence, we have $3^4 = 81$ many realizations of 4-model collections. We present the average performance as well as the standard deviation (*std*) of such 81 models' evaluations.

**Conventional evaluation.** We use Adam (Kingma & Ba, 2014) optimizer with default moment parameters, $\beta_1 = .9$ and $\beta_2 = .99$. Following recent works (Kim et al., 2020), we use *reduce on plateau* learning rate scheduler with patience 4. The initial learning rate is $10^{-5}$ for CUB, and $10^{-4}$ for Cars, SOP and In-shop. We use $10^{-4}$ weight decay for BNInception backbone and $4 \, 10^{-4}$ wight decay for ResNet50 backbone. We use batch size of 128 (4 samples per class) for BNInception backbone and 112 (4 samples per class) for ResNet backbone (following (Roth et al., 2020)). We evaluate validation P@1 for every 25 steps of training in CUB and Cars196, for 250 steps in SOP and In-shop. We stop training if no improvement is observed for 15 steps in CUB and Cars196, and 10 steps in SOP and In-shop. We recover the parameters with the best validation performance. We repeat each experiment 3 times and report the best result. For the evaluations on LIBC framework (Seidenschwarz et al., 2021), we follow their experimental setting.

**Hyperparameter tuning.** We use Adam (Kingma & Ba, 2014) optimizer with constant $10^{-5}$ learning rate, $10^{-4}$ weight decay, and default moment parameters, $\beta_1 = .9$ and $\beta_2 = .99$. We use batch size of 32 (4 samples per class). We evaluate validation MAP@R for every 100 steps of training in CUB and Cars196, for 1000 steps in SOP and In-shop. We stop training if no improvement is observed for 10 steps in CUB and Cars196, and 7 steps in SOP and In-shop. We recover the parameters with the best validation performance. We train a single model on the $4/5$ of the training set while using $1/5$ for the validation. We repeat each experiment 3 times and report the averaged result.

### 1.1.5 EMBEDDING VECTORS

**Fair evaluation.** Embedding dimension is fixed to 128. During training and evaluation, the embedding vectors are $\ell 2$ normalized. We follow the evaluation method proposed in (Musgrave et al., 2020) and produce two results: $i$) Average performance (128 dimensional) of 4-fold models and $ii$) Ensemble performance (concatenated 512 dimensional) of 4-fold models where the embedding vector is obtained by concatenated 128D vectors of the individual models before retrieval.

**Conventional evaluation.** Embedding dimension is 512 in BNInception and ResNet50 experiments for both XBM and LIBC.

**Hyperparameter tuning.** Embedding dimension is fixed to 128.

### 1.1.6 Baselines with GSP

We evaluate our method with *C1+XBM+GSP*: Cross-batch memory (XBM) (Wang et al., 2020) with original Contrastive loss (C1) (Hadsell et al., 2006), *C2+GSP*: Contrastive loss with positive margin (Wu et al., 2017), *MS+GSP*: Multi-similarity (MS) loss (Wang et al., 2019), *Triplet+GSP*: Triplet loss (Schroff et al., 2015), *PNCA+GSP*: ProxyNCA++ loss (Teh et al., 2020), *PAnchor+GSP*: ProxyAnchor loss (Kim et al., 2020).

### 1.1.7 Hyperparameters

For the hyperparameter selection, we exploit the recent work (Musgrave et al., 2020) that has performed parameter search via Bayesian optimization on variety of losses. We further experiment the suggested parameters from the original papers and official implementations. We pick the best performing parameters. We perform no further parameter tuning for the baseline methods' parameters when applied with our method to purely examine the effectiveness of our method.

**C1**: We adopted XBM's official implementation for fair comparison. We use 0.5 margin for all datasets.

**C2**: C2 has two parameters, $(m^+, m^-)$: positive margin, $m^+$, and negative margin. We set $(m^+, m^-)$ to $(0, 0.3841), (0.2652, 0.5409), (0.2858, 0.5130), (0.2858, 0.5130)$ for CUB, Cars196, In-shop and SOP, respectively.

**Triplet**: We set its margin to 0.0961, 0.1190, 0.0451, 0.0451 for CUB, Cars196, In-shop and SOP, respectively.

**MS**: MS has three parameters $(\alpha, \beta, \lambda)$. We set $(\alpha, \beta, \lambda)$ to $(2, 40, 0.5)$, $(14.35, 75.83, 0.66)$, $(8.49, 57.38, 0.41)$, $(2, 40, 0.5)$ for CUB, Cars196, In-shop and SOP, respectively.

**ProxyAnchor**: We set its two paremeters $(\delta, \alpha)$ to $(0.1, 32)$ for all datasets. We use 1 sample per class in batch setting (*i.e.*, 32 classes with 1 samples per batch), we perform 1 epoch warm-up training of the embedding layer, and we apply learning rate multiplier of 100 for the proxies during training. For SOP dataset, we use $5 \, 10^{-6}$ learning rate.

**ProxyNCA++**: We set its temperature parameter to 0.11 for all datasets. We use 1 sample per class in batch setting (*i.e.*, 32 classes with 1 samples per batch), we perform 1 epoch warm-up training of the embedding layer, and we apply learning rate multiplier of 100 for the proxies during training.

**XBM**: We evaluate XBM with C1 since in the original paper, contrastive loss is reported to be the best performing baseline with XBM. We set the memory size of XBM according to the dataset. For CUB and Cars196, we use memory size of 25 batches. For In-shop, we use 400 batches and for SOP we use 1400 batches. We perform 1K steps of training with the baseline loss prior to integrate XBM loss in order to ensure XBM's *slow drift* assumption.

**GSP**: For the hyperparameters of our method, we perform parameters search, details of which are provided in § 1.6. We use 64-many prototypes in CUB and Cars, and 128-many prototypes in SOP and In-shop. We use $\varepsilon = 0.5$ for proxy-based losses and $\varepsilon = 5.0$ for non-proxy losses. For the rest, we set $\mu = 0.3$, $\epsilon = 0.05$, and we iterate until $k = 100$ in Proposition 4.1. For zero-shot prediction loss coefficient (*i.e.*, $(1-\lambda)\mathcal{L}_{DML} + \lambda\mathcal{L}_{ZS}$), we set $\lambda = 0.1$.

## 1.2 Fair Evaluation

We compare our method (GSP) against direct application of GAP with 6 DML methods in 4 datasets. We also compare our method with generalized mean pooling (GeMean) (Radenović et al., 2018) and DeLF (Noh et al., 2017), based on the results of evaluation 13 additional pooling alternatives on *Ciffar Collage* and CUB datasets (Tab. 4). We also evaluate max pooling (GMP) and its combination with GAP as we typically observe GAP+GMP in the recent works (Venkataramanan et al., 2022; Teh et al., 2020; Kim et al., 2020; Wang et al., 2020). We also apply our method with GMP (GMP+GSP) and with GeMean (GeMean+GSP) to show that per channel selection is orthogonal to our approach and thus, GSP can marginally improve those methods as well.

We observe consistent improvements upon direct application of GAP in all datasets. Predominantly, we achieve more improvement in MAP@R metric than R@1 (P@1) which is shown to be a noisy

Table 2: Comparison with the existing methods for the retrieval task on SOP, In-shop, CUB, Cars datasets. Experimental setting follows § 1.1-*Fair evaluation*. ∓ denotes 1 *std* margin. Red: the best, Blue: the second best, Bold: the loss term specific best.

| Dataset → | SOP | | | | In-shop | | | | CUB | | | | Cars196 | | | |
|---|---|---|---|---|---|---|---|---|---|---|---|---|---|---|---|---|
| Dim. → | 512D | | 128D | | 512D | | 128D | | 512D | | 128D | | 512D | | 128D | |
| Method↓ | P@1 | MAP@R | P@1 | MAP@R | P@1 | MAP@R | P@1 | MAP@R | P@1 | MAP@R | P@1 | MAP@R | P@1 | MAP@R | P@1 | MAP@R |
| **C1+** | | | | | | | | | | | | | | | | |
| GAP | 69.29 ∓0.11 | 40.40 ∓0.15 | 65.15 ∓0.10 | 36.50 ∓0.11 | 80.11 ∓0.19 | 50.32 ∓0.14 | 75.83 ∓0.14 | 46.42 ∓0.13 | 63.32 ∓0.35 | 23.49 ∓0.31 | 56.34 ∓0.35 | 19.37 ∓0.29 | 78.01 ∓0.38 | 22.87 ∓0.33 | 65.61 ∓0.59 | 16.06 ∓0.14 |
| XBM+GAP | 76.54 ∓0.32 | 48.58 ∓0.47 | 73.22 ∓0.48 | 44.55 ∓0.57 | 87.76 ∓0.26 | 57.53 ∓0.41 | 85.26 ∓0.37 | 54.40 ∓0.45 | 65.56 ∓0.48 | 25.65 ∓0.24 | 57.48 ∓0.41 | 20.27 ∓0.19 | 83.55 ∓0.35 | 27.53 ∓0.22 | 72.17 ∓0.30 | 18.98 ∓0.17 |
| XBM+GSP | **77.88** ∓0.18 | **50.65** ∓0.28 | **74.84** ∓0.19 | **46.69** ∓0.28 | **88.33** ∓0.19 | **58.55** ∓0.29 | **85.95** ∓0.21 | **55.30** ∓0.21 | **67.00** ∓0.49 | **26.05** ∓0.15 | **58.89** ∓0.49 | **20.60** ∓0.16 | **83.31** ∓0.22 | **27.88** ∓0.23 | **73.04** ∓0.39 | **19.26** ∓0.19 |
| **C2+** | | | | | | | | | | | | | | | | |
| GAP | 74.20 ∓0.23 | 45.85 ∓0.31 | 70.54 ∓0.19 | 41.79 ∓0.26 | 86.47 ∓0.15 | 59.07 ∓0.21 | 83.42 ∓0.12 | 55.38 ∓0.13 | 67.35 ∓0.50 | 25.95 ∓0.21 | 58.87 ∓0.36 | 20.58 ∓0.13 | 80.96 ∓0.48 | 24.38 ∓0.58 | 69.55 ∓0.42 | 17.02 ∓0.31 |
| GSP | 74.91 ∓0.12 | 46.81 ∓0.17 | 71.43 ∓0.11 | 42.84 ∓0.14 | 86.90 ∓0.17 | 60.01 ∓0.29 | 83.57 ∓0.18 | 55.94 ∓0.17 | 68.85 ∓0.41 | 27.12 ∓0.27 | 60.42 ∓0.36 | 21.52 ∓0.16 | 82.83 ∓0.27 | 26.25 ∓0.34 | 71.40 ∓0.27 | 18.31 ∓0.22 |
| DeLF | 74.59 ∓0.15 | 46.54 ∓0.19 | 45.53 ∓0.18 | 42.47 ∓0.17 | 86.65 ∓0.16 | 59.20 ∓0.22 | 83.51 ∓0.09 | 55.36 ∓0.12 | 68.66 ∓0.32 | 27.06 ∓0.18 | 59.85 ∓0.18 | 21.42 ∓0.16 | 81.85 ∓0.41 | 24.77 ∓0.38 | 69.95 ∓0.38 | 17.32 ∓0.25 |
| GeMean | 74.92 ∓0.13 | 46.99 ∓0.11 | 71.53 ∓0.09 | 43.12 ∓0.12 | 86.62 ∓0.15 | 59.12 ∓0.19 | 83.83 ∓0.09 | 55.70 ∓0.12 | 68.79 ∓0.36 | 27.12 ∓0.20 | 60.37 ∓0.30 | 21.50 ∓0.15 | 82.43 ∓0.60 | 25.27 ∓0.63 | 70.23 ∓0.55 | 17.41 ∓0.45 |
| GeMean+GSP | **75.32** ∓0.08 | **47.69** ∓0.13 | **71.93** ∓0.10 | **43.71** ∓0.13 | **86.94** ∓0.15 | 59.98 ∓0.21 | **84.35** ∓0.19 | **56.34** ∓0.14 | **69.11** ∓0.34 | **27.56** ∓0.18 | **60.81** ∓0.34 | **21.84** ∓0.19 | **83.62** ∓0.36 | **26.98** ∓0.31 | **72.38** ∓0.28 | **19.05** ∓0.22 |
| GMP | 74.09 ∓0.15 | 46.13 ∓0.19 | 69.68 ∓0.20 | 41.31 ∓0.22 | 86.38 ∓0.12 | 59.04 ∓0.10 | 83.04 ∓0.13 | 54.89 ∓0.07 | 68.13 ∓0.18 | 26.43 ∓0.21 | 58.99 ∓0.34 | 20.66 ∓0.18 | 81.83 ∓0.62 | 25.11 ∓0.72 | 69.05 ∓0.61 | 17.08 ∓0.47 |
| GMP+GAP | 74.71 ∓0.11 | 46.70 ∓0.15 | 70.83 ∓0.10 | 42.38 ∓0.15 | 86.58 ∓0.16 | 59.22 ∓0.18 | 83.41 ∓0.12 | 55.37 ∓0.11 | 67.88 ∓0.23 | 26.63 ∓0.23 | 59.24 ∓0.32 | 20.88 ∓0.17 | 82.14 ∓0.40 | 25.66 ∓0.44 | 69.81 ∓0.38 | 17.62 ∓0.32 |
| GMP+GSP | 75.08 ∓0.1 | 47.12 ∓0.17 | 71.18 ∓0.15 | 42.80 ∓0.18 | 86.79 ∓0.16 | 59.43 ∓0.28 | 83.86 ∓0.15 | 55.76 ∓0.19 | 68.47 ∓0.58 | 27.49 ∓0.36 | 60.19 ∓0.41 | 21.69 ∓0.35 | 82.54 ∓0.46 | 26.30 ∓0.43 | 71.03 ∓0.48 | 18.24 ∓0.29 |
| **MS+** | | | | | | | | | | | | | | | | |
| GAP | 72.81 ∓0.14 | 44.19 ∓0.21 | 69.09 ∓0.10 | 40.34 ∓0.16 | 87.01 ∓0.20 | 58.79 ∓0.37 | 83.87 ∓0.21 | 54.85 ∓0.34 | 65.43 ∓0.46 | 24.95 ∓0.15 | **57.57** ∓0.27 | 20.13 ∓0.12 | 83.73 ∓0.34 | 27.16 ∓0.43 | 72.54 ∓0.43 | 18.73 ∓0.31 |
| GSP | **73.05** ∓0.11 | **44.72** ∓0.17 | **69.44** ∓0.15 | **40.87** ∓0.19 | **88.28** ∓0.21 | **60.49** ∓0.24 | **85.28** ∓0.19 | **56.62** ∓0.26 | 65.50 ∓0.33 | 25.09 ∓0.21 | 57.39 ∓0.15 | 20.34 ∓0.22 | **84.27** ∓0.35 | **28.58** ∓0.40 | **73.74** ∓0.32 | **19.91** ∓0.31 |
| **Triplet+** | | | | | | | | | | | | | | | | |
| GAP | 74.54 ∓0.24 | 45.88 ∓0.30 | 69.41 ∓0.38 | 40.01 ∓0.39 | 85.99 ∓0.36 | 59.67 ∓0.46 | 81.75 ∓0.38 | 54.25 ∓0.45 | 64.11 ∓0.66 | 23.65 ∓0.40 | 55.62 ∓0.46 | 18.54 ∓0.31 | 77.58 ∓0.60 | 22.67 ∓0.58 | 64.61 ∓0.59 | 15.74 ∓0.34 |
| GSP | **75.59** ∓0.23 | **47.35** ∓0.32 | **70.65** ∓0.20 | **41.38** ∓0.22 | **86.75** ∓0.27 | **60.85** ∓0.47 | **82.74** ∓0.33 | **55.54** ∓0.46 | **66.09** ∓0.52 | **24.80** ∓0.33 | **57.12** ∓0.42 | **19.38** ∓0.25 | **78.93** ∓0.30 | **23.44** ∓0.29 | **65.81** ∓0.35 | **16.14** ∓0.21 |
| **PNCA+** | | | | | | | | | | | | | | | | |
| GAP | 75.18 ∓0.15 | 47.11 ∓0.16 | 72.15 ∓0.06 | 43.57 ∓0.08 | 87.26 ∓0.14 | 57.43 ∓0.14 | 84.86 ∓0.08 | 54.41 ∓0.10 | 65.74 ∓0.51 | 25.27 ∓0.23 | 58.19 ∓0.36 | 20.63 ∓0.10 | 82.33 ∓0.25 | 26.21 ∓0.22 | 70.75 ∓0.18 | 18.61 ∓0.08 |
| GSP | 75.68 ∓0.11 | 47.74 ∓0.14 | 72.37 ∓0.06 | 43.95 ∓0.06 | 87.35 ∓0.10 | 57.65 ∓0.12 | 85.13 ∓0.10 | 54.68 ∓0.08 | 65.80 ∓0.25 | 25.48 ∓0.25 | 58.20 ∓0.25 | 20.75 ∓0.15 | 82.70 ∓0.27 | 26.93 ∓0.22 | 71.55 ∓0.32 | 19.20 ∓0.17 |
| DeLF | 75.29 ∓0.09 | 47.44 ∓0.11 | 72.05 ∓0.06 | 43.62 ∓0.07 | 87.19 ∓0.11 | 57.44 ∓0.10 | 84.55 ∓0.04 | 54.13 ∓0.06 | 65.42 ∓0.15 | 25.31 ∓0.16 | 57.98 ∓0.20 | 20.51 ∓0.14 | 82.37 ∓0.35 | 26.63 ∓0.27 | 71.06 ∓0.27 | 18.81 ∓0.14 |
| GeMean | 75.64 ∓0.09 | 47.82 ∓0.07 | 72.75 ∓0.07 | 44.43 ∓0.06 | 87.63 ∓0.10 | 57.88 ∓0.14 | 85.48 ∓0.12 | 55.14 ∓0.10 | 66.33 ∓0.33 | 25.74 ∓0.20 | 58.52 ∓0.36 | 20.71 ∓0.15 | **83.83** ∓0.29 | 27.44 ∓0.15 | **72.14** ∓0.28 | 19.16 ∓0.12 |
| GeMean+GSP | **75.89** ∓0.11 | **48.17** ∓0.12 | **72.91** ∓0.04 | **44.61** ∓0.06 | **87.64** ∓0.10 | **58.12** ∓0.16 | **85.58** ∓0.07 | **55.25** ∓0.08 | **67.39** ∓0.53 | **26.19** ∓0.26 | **59.39** ∓0.40 | **21.31** ∓0.21 | 83.09 ∓0.25 | **27.96** ∓0.30 | 71.95 ∓0.27 | **19.74** ∓0.19 |
| GMP | 74.43 ∓0.08 | 46.33 ∓0.08 | 70.80 ∓0.07 | 42.24 ∓0.08 | 86.94 ∓0.13 | 56.79 ∓0.13 | 84.53 ∓0.08 | 53.86 ∓0.09 | 65.51 ∓0.51 | 25.36 ∓0.29 | 57.61 ∓0.38 | 20.33 ∓0.29 | 83.06 ∓0.33 | 26.96 ∓0.27 | 71.19 ∓0.25 | 18.92 ∓0.15 |
| GMP+GAP | 75.19 ∓0.09 | 47.26 ∓0.11 | 71.97 ∓0.04 | 43.55 ∓0.06 | 87.21 ∓0.14 | 57.34 ∓0.15 | 84.95 ∓0.09 | 54.42 ∓0.10 | 65.91 ∓0.35 | 25.56 ∓0.26 | 57.92 ∓0.37 | 20.68 ∓0.20 | 82.92 ∓0.41 | 26.92 ∓0.36 | 71.33 ∓0.22 | 18.95 ∓0.19 |
| GMP+GSP | 75.41 ∓0.12 | 47.50 ∓0.12 | 72.10 ∓0.07 | 43.73 ∓0.09 | 87.43 ∓0.10 | 57.68 ∓0.14 | 85.10 ∓0.10 | 54.70 ∓0.08 | 66.14 ∓0.48 | 25.85 ∓0.23 | 58.12 ∓0.32 | 20.96 ∓0.20 | 83.46 ∓0.31 | 27.12 ∓0.21 | 72.04 ∓0.39 | 19.38 ∓0.20 |
| **PAnchor+** | | | | | | | | | | | | | | | | |
| GAP | 76.48 ∓0.19 | 48.08 ∓0.26 | 73.50 ∓0.14 | 44.33 ∓0.20 | 88.02 ∓0.21 | 58.02 ∓0.25 | 85.83 ∓0.18 | 54.98 ∓0.22 | 68.04 ∓0.41 | 26.20 ∓0.21 | 59.91 ∓0.34 | 20.94 ∓0.15 | 85.26 ∓0.31 | 27.14 ∓0.20 | 75.08 ∓0.23 | 19.15 ∓0.13 |
| GSP | 77.13 ∓0.16 | 49.05 ∓0.22 | 74.07 ∓0.13 | 45.07 ∓0.17 | 88.10 ∓0.11 | 58.44 ∓0.14 | 85.97 ∓0.06 | 55.34 ∓0.13 | 68.40 ∓0.45 | 26.59 ∓0.25 | 60.80 ∓0.31 | 21.44 ∓0.17 | 86.46 ∓0.39 | 28.43 ∓0.33 | 75.88 ∓0.25 | 19.90 ∓0.20 |

measure for DML evaluation (Musgrave et al., 2020). On the average, we consistently improve the baselines ≈1% points in MAP@R. Our improvement margins are superior to ones of attention based DeLF pooling. We improve state-of-the-art (SOTA) XBM method up to 2% points, which is a good evidence that application of GSP is not limited to loss terms but can be combined with different DML approaches. We also consistently improve GMP and GeMean pooling methods in all datasets, yet another evidence that our method can be combined with max pooling based methods.

## 1.3 CONVENTIONAL EVALUATION

We additionally follow the relatively old-fashioned conventional procedure (Oh Song et al., 2016) for the evaluation of our method. We use BN-Inception (Ioffe & Szegedy, 2015) and ResNet50 (He et al., 2016) architectures as the backbones. We obtain 512D (BN-Inception and ResNet50) embeddings through linear transformation after global pooling layer. Aligned with the recent approaches (Venkataramanan et al., 2022; Teh et al., 2020; Kim et al., 2020; Wang et al., 2020), we use global max pooling as well as global average pooling. The rest of the settings are disclosed in § 1.1.

Table 3: Comparison with the existing methods for the retrieval task in conventional experimental settings with BN-Inception and ResNet50 backbones where superscripts denote embedding size. Red: the best. Blue: the second best. Bold: previous SOTA. [†]*Results obtained from* (Seidenschwarz et al., 2021).

(a)

| Backbone → | BN-Inception-512D | | | |
|---|---|---|---|---|
| Dataset → | CUB | Cars196 | SOP | In-shop |
| Method ↓ | R@1 | R@1 | R@1 | R@1 |
| C1+XBM (Wang et al., 2020) | 65.80 | 82.00 | 79.50 | 89.90 |
| ProxyAnchor (Kim et al., 2020) | 68.40 | 86.10 | 79.10 | 91.50 |
| DiVA (Milbich et al., 2020) | 66.80 | 84.10 | 78.10 | - |
| ProxyFewer (Zhu et al., 2020) | 66.60 | 85.50 | 78.00 | - |
| Margin+S2SD (Roth et al., 2021) | 68.50 | 87.30 | 79.30 | - |
| C1+XBM | 64.32 (23.59) | 77.63 (21.67) | 79.29 (52.59) | 90.16 (61.39) |
| C1+XBM+GSP | 64.99 (25.35) | 79.07 (22.51) | 79.59 (52.70) | 90.92 (63.25) |

(b)

| Backbone → | ResNet50 | | | |
|---|---|---|---|---|
| Dataset → | CUB | Cars196 | SOP | In-shop |
| Method ↓ | R@1 | R@1 | R@1 | R@1 |
| C1+XBM[128] (Wang et al., 2020) | - | - | 80.60 | 91.30 |
| ProxyAnchor[512] (Kim et al., 2020) | 69.70 | 87.70 | 80.00[†] | 92.10[†] |
| DiVA[512] (Milbich et al., 2020) | 69.20 | 87.60 | 79.60 | - |
| ProxyNCA++[512] (Teh et al., 2020) | 66.30 | 85.40 | 80.20 | 88.60 |
| Margin+S2SD[512] (Roth et al., 2021) | 69.00 | 89.50 | 81.20 | - |
| LIBC[512] (Seidenschwarz et al., 2021) | 70.30 | 88.10 | 81.40 | 92.80 |
| MS+Metrix[512] (Venkataramanan et al., 2022) | 71.40 | 89.60 | 81.00 | 92.20 |
| PAnchor+DIML[128] (Zhao et al., 2021) | 66.46 (25.58) | 86.13 (28.11) | 79.22 (43.04) | - |
| LIBC+GSP[512] | 70.70 | 88.43 | 81.65 | 93.30 |
| C1+XBM[512] | 66.68 (25.38) | 82.83 (25.34) | 81.44 (55.66) | 91.56 (64.00) |
| C1+XBM+GSP[512] | 66.63 (25.51) | 82.60 (25.76) | 81.54 (55.91) | 91.75 (64.43) |

We evaluate our method with XBM. We provide R@1 results in Tab. 3 for the comparison with SOTA. In our evaluations, we also provide MAP@R scores in parenthesis under R@1 scores. We also provide baseline XBM evaluation in our framework. The results are mostly consistent with the ones reported in the original paper (Wang et al., 2020) except for CUB and Cars datasets. In XBM (Wang et al., 2020), the authors use proxy-based trainable memory for CUB and Cars datasets. On the other hand, we use the official implementation provided by the authors, which does not include such proxy-based extensions.

We observe that our method improves XBM and XBM+GSP reaches SOTA performance in large scale datasets. With that being said, the improvement margins are less substantial than the ones in *fair evaluation*. Such a result is expected since training is terminated by *early-stopping* which is a common practice to regularize the generalization of training (Dong et al., 2020; Lei et al., 2021). In *conventional evaluation*, early-stopping is achieved by monitoring the test data performance, enabling good generalization to test data. Therefore, observing less improvement in generalization with GSP is something we expect owing to generalization boost that test data based early-stopping already provides.

We also observe that in a few cases, the R@1 performance of GSP is slightly worse than the baseline. Nevertheless, once we compare the MAP@R performances, GSP consistently brings improvement with no exception. We should recapitulate that R@1 is a myopic metric to assess the quality of the embedding space geometry (Musgrave et al., 2020) and hence, pushing R@1 does not necessarily reflect the true order of the improvements that the methods bring.

As we observe from MAP@R comparisons in Table 2, the methods sharing similar R@1 (*i.e.*, P@1) performances can differ in MAP@R performance relatively more significantly. In that manner, we firmly believe that comparing MAP@R performances instead of R@1 technically sounds more in showing the improvements of our method.

Finally, we also apply our method with LIBC (Seidenschwarz et al., 2021) to further show wide applicability of our method. We use the official implementation of LIBC and follow their default experimental settings. The evaluations on 4 benchmarks show that GSP improve LIBC by $\approx$ 0.5pp R@1.

Table 4: Evaluation of feature pooling methods on *Cifar Collage* and CUB datasets with Contrastive and ProxyNCA++ losses for DML task. Red: the best, Blue: the second best, Bold: the third best.

| | 128D - MAP@R | | | |
|---|---|---|---|---|
| Dataset→ | **Cifar Collage** | | **CUB** | |
| **Method↓ Loss→** | **Contrastive** | **ProxyNCA++** | **Contrastive** | **ProxyNCA++** |
| CBAM (Woo et al., 2018) | 7.87 | 10.99 | 18.45 | 18.21 |
| CroW (Kalantidis et al., 2016) | 10.09 | 11.48 | 20.88 | 20.42 |
| DeLF (Noh et al., 2017) | 11.44 | 24.83 | **21.42** | 20.51 |
| GeMax Murray et al. (2016) | 7.04 | 7.83 | 18.85 | 17.83 |
| GeMean Radenović et al. (2018) | 10.97 | 10.60 | 21.50 | 20.71 |
| GSoP (Gao et al., 2019) | 11.15 | 17.73 | 20.52 | 15.72 |
| OTP (Mialon et al., 2021) 8×16|64×128 | 7.02 | 11.55 | 15.19 \| 20.88 | 13.57 \| 20.48 |
| SOLAR (Ng et al., 2020) | **17.30** | 20.36 | 19.89 | 20.14 |
| T-SMK (Tolias et al., 2020) | 9.21 | 13.15 | 21.01 | 20.23 |
| VLAD (Arandjelovic et al., 2016) 8×16|64×128 | 21.73 | 19.68 | 15.19 \| 16.67 | 13.08 \| 16.53 |
| WELDON (Durand et al., 2016) | 13.81 | **20.38** | 20.67 | 20.31 |
| GAP | 8.09 | 10.68 | 20.58 | 20.63 |
| GMP | 9.53 | 11.25 | 20.66 | 20.33 |
| GMP+GAP | 10.01 | 11.85 | 20.88 | **20.68** |
| GSP | **22.68** | **27.61** | **21.52** | **20.75** |

## 1.4 EVALUATION OF OTHER POOLING ALTERNATIVES

We evaluate 13 additional pooling alternatives on *Ciffar Collage* and CUB datasets with *contrastive* (C2) and *Proxy-NCA++* (PNCA) losses. We pick *contrastive* since it is one of the best performing sample-based loss. We pick *Proxy-NCA++* since most of the pooling methods are tailored for landmark-based image retrieval and use classification loss akin to *Proxy-NCA++*. We particularly consider *Cifar Collage* dataset since the images of different classes share a considerable amount of semantic entities, enabling us to assess the methods with respect to their ability to discard the nuisance information.

In addition to our method (GSP) and global average pooling (GAP), we consider: $i$) global max pooling (GMP), $ii$) GAP+GMP (Kim et al., 2020), $iii$) CBAM (Woo et al., 2018), $iv$) CroW (Kalantidis et al., 2016), $v$) DeLF (Noh et al., 2017), $vi$) generalized max pooling (GeMax) (Murray et al., 2016), $vii$) generalized mean pooling (GeMean) (Radenović et al., 2018), $viii$) GSoP (Gao et al., 2019), $ix$) optimal transport based aggregation (OTP) (Mialon et al., 2021), $x$) SOLAR (Ng et al., 2020), $xi$) trainable SMK (T-SMK) (Tolias et al., 2020), $xii$) NetVLAD (Arandjelovic et al., 2016), and $xiii$) WELDON (Durand et al., 2016). Among those, OTP and VLAD are ensemble based methods and typically necessitate large embedding dimensions. Thus, we both experimented their 128 dimensional version -(8×16) (8 prototypes of 16 dimensional vectors) and 8192 dimensional version -(64×128) (64 prototypes of 128 dimensional vectors).

For CUB dataset, the experimental setting follows § 1.1-*Fair evaluation* and we report MAP@R performance of the 4-model average at 128 dimensional embeddings each. For *Cifar Collage dataset*, the experimental setting follows § 2.2 and we report MAP@R performance. We provide the results in Tab. 4.

Evaluations show that our method is superior to other pooling alternatives including prototype based VLAD and OTP. Predominantly, for 128 dimensional embeddings, our method outperforms prototype based methods by large margin. In CUB dataset, the pooling methods either are inferior to or perform on par with GAP. The performance improvements of the superior methods are less than 1%, implying that our improvements in the order of 1-2% reported in Table 2 is substantial. On the other hand, the methods that mask the feature map outperform GAP by large margin in *Cifar Collage* dataset. That being said, our method outperforms all the methods except for Contrastive+VLAD by large margin in *Cifar Collage* dataset, yet another evidence for better feature selection mechanism of our method.

For instance in CUB dataset, DeLF and GeMean are on par with our method which has slightly better performance. Yet, our method outperforms both methods by large margin in *Cifar Collage* dataset.

Comparing to CroW, T-SMK and CBAM, our method outperforms those methods by large margin. Those methods are the built on feature magnitude based saliency, assuming that the backbone functions must be able to zero-out nuisance information. Yet, such a requirement is restrictive for the parameter space and annihilation of the non-discriminative information might not be feasible in some problems. We experimentally observe such a weakness of CroW, T-SMK and CBAM in *Cifar Collage* dataset where the nuisance information cannot be zeroed-out by the backbone. Our formulation do not have such a restrictive assumption and thus substantially superior to those methods.

Similarly, attention-based weighting methods, DeLF and GSoP, do not have explicit control on feature selection behavior and might result in poor models when jointly trained with the feature extractor (Noh et al., 2017), which we also observe in *Cifar Collage* experiments. On the contrary, we have explicit control on the pooling behavior with $\mu$ parameter and the behavior of our method is stable and consistent across datasets and with different loss functions.

Moreover, attention-based methods DeLF, GSoP, and SOLAR typically introduce several convolution layers to compute the feature weights. We only introduce an $m$-kernel 1x1 convolution layer (*i.e.*, $m$-many trainable prototypes) and obtain better results. We should note that our pooling operation is as simple as performing a convolution (*i.e.*, distance computation) and alternating normalization of a vector and a scalar. That being said, we are able to incorporate a zero-shot regularization loss into our problem naturally by using the prototype assignment weights. We can as well incorporate such a loss in DeLF which has 1x1 convolution to compute

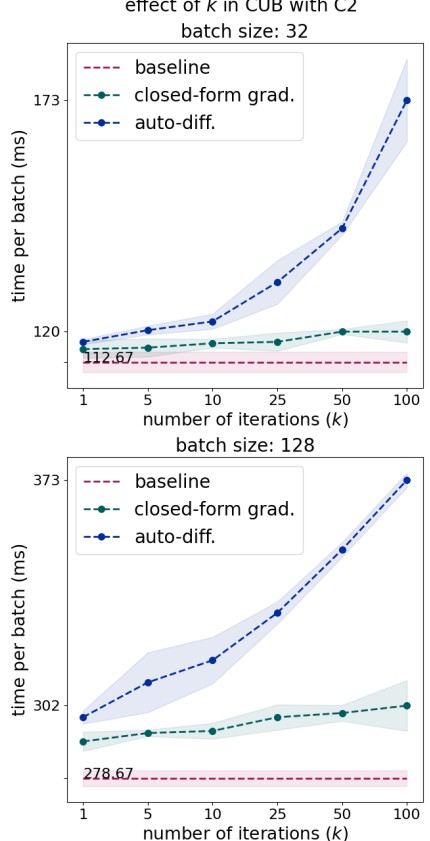

prototype similarities. However, we first need a mechanism to aggregate the per prototype similarities (*e.g.* sum and normalization). Indeed, normalizing the similarities channel-wise and spatially summing them correspond to solving our problem with $\mu = 1$.

Other pooling methods, *i.e.*, GAP, GMP, GAP+GMP, GeMax, GeMean, WELDON, VLAD, OTP, do not build on discriminative feature selection. Thus, our method substantially outperforms those.

## 1.5 Computational Analysis

Forward and backward computation of proposed GSP method can be implemented using only matrix-vector products. Moreover, having closed-form matrix-inversion-free expression for the loss gradient enables memory efficient back propagation since the output of each iteration must be stored otherwise.

We perform $k$ iterations to obtain the pooling weights and at each iteration, we only perform matrix-vector products. In this sense, the back propagation can be achieved using *automatic-differentiation*. One problem with automatic differentiation is that the computation load increases with increasing $k$. On the other hand, with closed-form gradient expression, we do not have such issue and in fact we have constant back propagation complexity. Granted that the closed-form expression demands exact solution of the problem (*i.e.*, $k \to \infty$), forward computation puts a little computation overhead and is memory efficient since we discard the intermediate outputs. Moreover, our initial empirical study show that our problems typically converges for $k > 50$ and we observe similar performances with $k \geqslant 25$.

Figure 7: Comparing closed-form gradient with automatic differentiation through analyzing the effect of $k$ on computation in CUB dataset with C2 loss. Shaded regions represent $\mp std$.

The choice of $k$ is indeed problem dependent (*i.e.*, size of the feature map and number of prototypes). Thus, it is important to see the effect of $k$ on computation load. We analyze the effect of $k$ with automatic differentiation and with our closed-form gradient expression. We provide the related plots in Fig. 7. We observe that with closed-form gradients, our method puts a little computation overhead and increasing $k$ has marginal effect. On the contrary, with automatic differentiation, the computational complexity substantially increases.

## 1.6 HYPERPARAMETER SELECTION

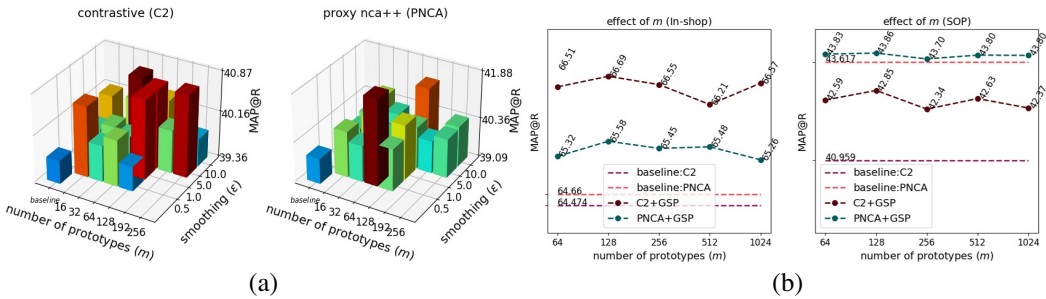

Figure 8: Parameter search for $m$ : number of prototoypes and $\varepsilon$: entropy smoothing coefficient. We fix $\mu = 0.3$ and $\lambda = 0.1$. (a) Searching $m - \varepsilon$ space in CUB dataset. (b) Effect of $m$ once we fix $\varepsilon = 5$ for Contrastive (C2) and $\varepsilon = 0.5$ for Proxy NCA++ (PNCA).

We first perform a screening experiment to see the effect of the parameters. We design a 2-level fractional factorial (*i.e.*, a subset of the all possible combinations) experiment. We provide the results in Tab. 5. In our analysis, we find that *lower the better* for $\lambda$ and $\mu$. Thus, we set $\mu = 0.3$ and $\lambda = 0.1$. $\varepsilon$ is observed to have the most effect and number of prototypes, $m$, seems to have no significant effect. Nevertheless, we jointly search for $m$ and $\varepsilon$. To this end, we perform grid search in CUB dataset with Contrastive (C2) and Proxy NCA++ (PNCA) losses. We provide the results in Fig. 8-(a). We see that both losses have their best performance when $m = 64$. On the other hand, $\varepsilon = 5.0$ works better for C2 while $\varepsilon = 0.5$ works better for PNCA. We additionally perform a small experiment to see whether $\varepsilon = 0.5$ is the case for Proxy Anchor loss as well and observe that $\varepsilon = 0.5$ is a better choice over $\varepsilon = 5.0$. As the result of $m$-$\varepsilon$ search, we set $\varepsilon = 5.0$ for non-proxy based losses and $\varepsilon = 0.5$ for proxy-based losses.

Table 5: Initial 2-level fractional factorial screening experiments for hyperparameter tuning (conducted in CUB).

| Setting | | | | MAP@R | |
|---|---|---|---|---|---|
| $m$ | $\mu$ | $\varepsilon$ | $\lambda$ | C2 | PNCA |
| 16 | 0.3 | 0.5 | 0.1 | 40.63 | 40.59 |
| 16 | 0.7 | 0.5 | 0.5 | 40.41 | 40.34 |
| 128 | 0.3 | 0.5 | 0.5 | 40.22 | 40.35 |
| 128 | 0.7 | 0.5 | 0.1 | 40.07 | 40.85 |
| 16 | 0.3 | 20 | 0.5 | 36.11 | 40.51 |
| 16 | 0.7 | 20 | 0.1 | 39.11 | 39.88 |
| 128 | 0.3 | 20 | 0.1 | 39.61 | 39.12 |
| 128 | 0.7 | 20 | 0.5 | 35.36 | 39.92 |
| Baseline | | | | 39.77 | 39.90 |

Fixing $\mu = 0.3, \lambda = 0.1, \varepsilon = 0.5$( or 5.0), we further experiment the effect of number of prototypes, $m$, in large datasets (*i.e.*, SOP and In-shop). We provide the corresponding performance plots in Fig. 8-(b). Supporting our initial analysis, $m$ seemingly does not have a significant effect once it is not small (*e.g.* $m \geqslant 64$). We observe that any choice of $m \geqslant 64$ provides performance improvement. With that being said, increasing $m$ does not bring substantial improvement over relatively smaller values. Considering the results of the experiment, we set $m = 128$ for SOP and In-shop datasets since both C2 and PNCA losses perform slightly better with $m = 128$ than the other choices of $m$.

## 2 DETAILS OF THE OTHER EMPIRICAL WORK

### 2.1 SYNTHETIC STUDY

We design a synthetic empirical study to evaluate GSP in a fully controlled manner. We consider 16-class problem such that classes are defined over trainable tokens. In this setting, tokens correspond to semantic entities but we choose to give a specific working to emphasize that they are trained as

part of the learning. Each class is defined with 4 distinct tokens and there are also 4 background tokens shared by all classes. For example, a *"car"* class would have tokens like *"tire"* and *"window"* as well as background tokens of *"tree"* and *"road"*.

We sample class representations from both class specific and background tokens according to a mixing ratio $\tilde{\mu} \sim \mathcal{N}(0.5, 0.1)$. We sample a total of 50 tokens and such a 50-many feature collection will correspond to a training sample (*i.e.*, we are mimicking CNN's output with trainable tokens). For instance, given class tokens for class-$c$, $\nu^{(c)} = \{\nu_1^{(c)}, \nu_2^{(c)}, \nu_3^{(c)}, \nu_4^{(c)}\}$ and shared tokens, $\nu^{(b)} = \{\nu_1^{(b)}, \nu_2^{(b)}, \nu_3^{(b)}, \nu_4^{(b)}\}$; we first sample $\mu = 0.4$ and then sample 20 tokens from $\nu^{(c)}$ with replacement, and 30 tokens from $\nu^{(b)}$, forming a feature collection for a class-$c$, *i.e.*, $f^{(c)} = \{\nu_3^{(c)}, \nu_1^{(c)}, \nu_1^{(c)}, \nu_3^{(c)}, \dots, \nu_4^{(b)}, \nu_3^{(b)}, \nu_4^{(b)}, \nu_1^{(b)}, \dots\}$ We then obtain global representations using GAP and GSP.

We do not apply $\ell 2$ normalization on the global representations. We also constrain the range of the token vectors to be in between $[-0.3, 0.3]$ to bound the magnitude of the learned vectors. We use default Adam optimizer with $10^{-4}$ learning rate and perform early stopping with 30 epoch patience by monitoring MAP@R. In each batch, we use 4 samples from 16 classes.

## 2.2 CIFAR COLLAGE

We consider the 20 *super-classes* of Cifar100 dataset (Krizhevsky & Hinton, 2009) where each has 5 sub-classes. For each super-class, we split the sub-classes for train (2), validation (1), and test (2). We consider 4 super-classes as the shared classes and compose 4x4-stitched collage images for the rest 16 classes. In particular, we sample an image from a class and then sample 3 images from shared classes. We illustrate a sample formation process in Fig. 9.

We should note that the classes exploited in training, validation and test are disjoint. For instance, if a *tree* class is used as a shared class in training, then that *tree* class does not exist in validation or test set as a shared feature. Namely, in our problem setting, both the background and the foreground classes are disjoint across training, validation and test sets. Such a setting is useful to analyze zero-shot transfer capability of our method.

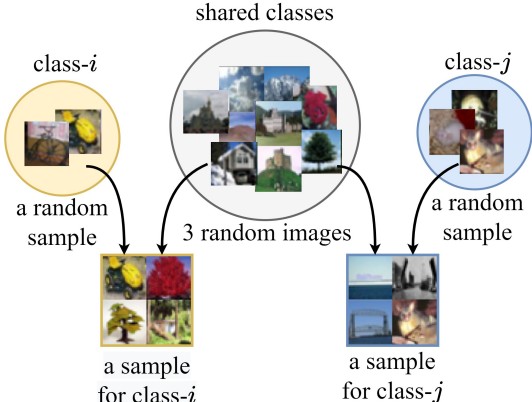

Figure 9: Illustration of a sample generation for Cifar Collage dataset.

We use ResNet20 (*i.e.*, 3 stages, 3 blocks) backbone pretrained on Cifar100 classification task. We use $\ell 2$ normalization on global representations. We use default Adam optimizer with initial 0.001 learning rate. We use *reduce on plateau* with 0.5 decay factor and 5 epochs patience. For GSP, we set $m = 128, \mu = 0.2, \varepsilon = 10, \lambda = 0.5$. We use 4 samples from 16 classes in a batch.

## 2.3 EVALUATION OF ZERO-SHOT PREDICTION LOSS

We train on Cifar10 (Krizhevsky & Hinton, 2009) dataset with 8 prototypes using ProxyNCA++ (Teh et al., 2020) (PNCA) loss with and without $\mathcal{L}_{ZS}$. We then use test set to compute pseudo-attribute histograms for each class. Namely, we aggregate the marginal transport plans of each sample in a class to obtain the histogram. Similarly, for each class, we compute the mean embedding vector (*i.e.*, we average embedding vectors of the samples of a class). Our aim is to fit a linear predictor to map attribute vectors to the mean embeddings.

To quantify the zero-shot prediction performance, we randomly split the classes into half and apply cross-batch zero-shot prediction. Specifically, we fit a linear predictor using 5 classes and then use that transformation to map the other 5 classes to their mean embeddings. We then compute pairwise distance between the predicted means and the true means. We then evaluate the nearest neighbour classification performance. We use both $\ell 2$ distance and cosine distance while computing the pairwise distances. We repeat the experiment 1000 times with different class splits.

## A APPENDIX

### A.1 PROOF FOR CLAIM 4.1

*Proof.* $\rho^*$ is obtained as the solution of the following optimal transport problem:

$$\rho^*, \pi^* = \underset{\substack{\rho, \pi \geqslant 0 \\ \rho_j + \Sigma_i \pi_{ij} = 1/n \\ \Sigma_{ij} \pi_{ij} = \mu}}{\arg\min} \sum_{ij} c_{ij} \pi_{ij}.$$

Given solutions $(\rho^*, \pi^*)$, for $\mu=1$, from the $3^{rd}$ constraint, we have $\Sigma_{ij}\pi_{ij}^*=1$. Then, using the $2^{nd}$ constraint, we write:

$$\sum_j \rho_j^* + \sum_j \sum_i \pi_{ij}^* = \sum_j \frac{1}{n}$$

where $j\in[n]$ for $n$-many convolutional features. Hence, we have $\sum_j \rho^* = 0$ which implies $\rho^*=0$ owing to non-negativity constraint. Finally, pooling weights becomes $p_i = \frac{1/n - \cancel{\rho_i}}{\underset{=1}{\mu}} = 1/n$.

$\square$

### A.2 PROOF FOR PROPOSITION 4.1

Before starting our proof, we first derive an iterative approach for the solution of (P2). We then prove that the iterations in Proposition 4.1 can be used to obtain the solution.

We can write (P2) equivalently as:

$$\rho^{(\varepsilon)}, \pi^{(\varepsilon)} = \underset{\substack{\rho, \pi \geqslant 0 \\ \rho_j + \Sigma_i \pi_{ij} = 1/n \\ \Sigma_{ij} \pi_{ij} = \mu}}{\arg\min} \sum_{ij} c_{ij}\pi_{ij} + \frac{1}{\varepsilon}\left(\sum_{ij} \pi_{ij}\log\pi_{ij} + \sum_j \rho_j \log\rho_j\right)$$

$$+ \sum_j 0\rho_j - \sum_{ij}\pi_{ij} - \sum_j \rho_j + \sum_{ij} e^{-\varepsilon c_{ij}} + \sum_j e^{-\varepsilon 0}$$

Rearranging the terms we get:

$$\rho^{(\varepsilon)}, \pi^{(\varepsilon)} = \underset{\substack{\rho, \pi \geqslant 0 \\ \rho_j + \Sigma_i \pi_{ij} = 1/n \\ \Sigma_{ij} \pi_{ij} = \mu}}{\arg\min} \sum_{ij} \pi_{ij}\log\frac{\pi_{ij}}{e^{-\varepsilon c_{ij}}} + \sum_j \rho_j\log\frac{\rho_j}{e^{-\varepsilon 0}} - \sum_{ij}\pi_{ij} - \sum_j \rho_j + \sum_{ij} e^{-\varepsilon c_{ij}} + \sum_j e^{-\varepsilon 0}$$

which is generalized *Kullback–Leibler divergence* (KLD) between $(\rho, \pi)$ and $(\exp(-\varepsilon 0), \exp(-\varepsilon c))$. Hence, we reformulate the problem as a KLD prjoection onto a convex set, which can be solved by *Bregman Projections* (*i.e.*, alternating projections onto constraint sets) (Bregman, 1967; Bauschke & Lewis, 2000). Defining $\mathcal{C}_1 := \{(\rho, \pi) \mid \rho_j + \sum_{ij} \pi_{ij} = 1/n \; \forall j\}$ and $\mathcal{C}_2 := \{(\rho, \pi) \mid \sum_{ij} \pi_{ij} = \mu\}$, and omitting constants, we can write the problem as:

$$\rho^{(\varepsilon)}, \pi^{(\varepsilon)} = \underset{\substack{\rho, \pi \geqslant 0 \\ (\rho, \pi) \in \mathcal{C}_1 \cap \mathcal{C}_2}}{\arg\min} \sum_{ij} \pi_{ij}\left(\log\frac{\pi_{ij}}{e^{-\varepsilon c_{ij}}} - 1\right) + \sum_j \rho_j\left(\log\frac{\rho_j}{e^{-\varepsilon 0}} - 1\right) \tag{P2'}$$

Given, $(\rho^{(k)}, \pi^{(k)})$, at iteration $k$, KLD projection onto $\mathcal{C}_1$, *i.e.*, $(\rho^{(k+1)}, \pi^{(k+1)}) := \mathcal{P}_{\mathcal{C}_1}^{KL}(\rho^{(k)}, \pi^{(k)})$, reads:

$$\rho_j^{(k+1)} = 1/n(\rho_j^{(k)} + \sum_i \pi_{ij}^{(k)})^{-1}\rho_j^{(k)},$$
$$\pi^{(k+1)} = 1/n(\rho_j^{(k)} + \sum_i \pi_{ij}^{(k)})^{-1}\pi_{ij}^{(k)}$$

where the results follow from *method of Lagrange multipliers*. Similarly, for $\mathcal{P}_{\mathcal{C}_2}^{KL}(\rho^{(k)}, \pi^{(k)})$, we have:

$$\rho^{(k+1)} = \rho^{(k)}, \quad \pi^{(k+1)} = \frac{\mu}{\sum_{ij}\pi_{ij}^{(k)}}\pi^{(k)}.$$

Given initialization, $(\rho^{(0)}, \pi^{(0)}) = (\mathbf{1}_n, \exp(-\varepsilon c))$, we obtain the pairs $(\rho^{(k)}, \pi^{(k)})$ for $k = 0, 1, 2, \ldots$ as:

$$\rho^{(k+1)} = 1/n(\rho^{(k)} + \pi^{(k)\intercal}\mathbf{1}_m)^{-1} \odot \rho^{(k)}, \quad \pi^{(k+1)} = \mu(\mathbf{1}_m^\intercal\hat{\pi}\mathbf{1}_n)^{-1}\hat{\pi}$$
$$\text{where} \quad \hat{\pi} = \pi^{(k)}Diag\left(1/n(\rho^{(k)} + \pi^{(k)\intercal}\mathbf{1}_m)^{-1}\right) \tag{A.1}$$

*Proof.* We will prove by induction. From Proposition 4.1, we have

$$\rho^{(k+1)} = 1/n\,(1 + t^{(k)}\exp(\text{-}\varepsilon c)^{\mathsf{T}}\mathbf{1}_m)^{\text{-}1}, \ \ t^{(k+1)} = \mu\,(\mathbf{1}_m^{\mathsf{T}}\exp(\text{-}\varepsilon c)\rho^{(k+1)})^{\text{-}1}$$

and $\pi^{(k)} = t^{(k)}\exp(\text{-}\varepsilon c)Diag(\rho^{(k)})$. It is easy to show that the expressions hold for the pair $(\rho^{(1)}, \pi^{(1)})$. Now, assuming that the expressions also holds for arbitrary $(\rho^{(k')}, \pi^{(k')})$. We have

$$\rho^{(k'+1)} = 1/n(\rho^{(k')} + \pi^{(k')\mathsf{T}}\mathbf{1}_m)^{\text{-}1} \odot \rho^{(k')}$$

Replacing $\pi^{(k')} = t^{(k')}\exp(\text{-}\varepsilon c)Diag(\rho^{(k')})$ we get:

$$\rho^{(k'+1)} = 1/n(\rho^{(k')} + Diag(\rho^{(k')})t^{(k')}\exp(\text{-}\varepsilon c)^{\mathsf{T}}\mathbf{1}_m)^{\text{-}1} \odot \rho^{(k')}$$

where $\rho^{(k')}$ terms cancel out, resulting in:

$$\rho^{(k'+1)} = 1/n(1 + t^{(k')}\exp(\text{-}\varepsilon c)^{\mathsf{T}}\mathbf{1}_m)^{\text{-}1}.$$

Similarly, we express $\hat{\pi}$ as:

$$\hat{\pi} = t^{(k')}\exp(\text{-}\varepsilon c)Diag(\rho^{k'})Diag\left(1/n\big(\rho^{(k')} + Diag(\rho^{(k')})t^{(k')}\exp(\text{-}\varepsilon c)^{\mathsf{T}}\mathbf{1}_m\big)^{\text{-}1}\right)$$

again $\rho^{(k')}$ terms cancel out, resulting in:

$$\hat{\pi} = t^{(k')}\exp(\text{-}\varepsilon c)Diag(1/n(1 + t^{(k')}\exp(\text{-}\varepsilon c)^{\mathsf{T}}\mathbf{1}_m)^{\text{-}1}) = t^{(k')}\exp(\text{-}\varepsilon c)Diag(\rho^{(k'+1)}).$$

Hence, $\pi^{(k'+1)}$ becomes:

$$\begin{aligned}
\pi^{(k'+1)} &= \mu(\mathbf{1}_m^{\mathsf{T}}t^{(k')}\exp(\text{-}\varepsilon c)Diag(\rho^{(k'+1)})\mathbf{1}_n)^{\text{-}1}t^{(k')}\exp(\text{-}\varepsilon c)Diag(\rho^{(k'+1)}) \\
&= \tfrac{1}{t^{(k')}}\underbrace{\mu(\mathbf{1}_m^{\mathsf{T}}\exp(\text{-}\varepsilon c)\rho^{(k'+1)})^{\text{-}1}}_{=t^{(k'+1)}}t^{(k')}\exp(\text{-}\varepsilon c)Diag(\rho^{(k'+1)}) \\
&= t^{(k'+1)}\exp(\text{-}\varepsilon c)Diag(\rho^{(k'+1)}),
\end{aligned}$$

meaning that the expressions also hold for the pair $(\rho^{(k'+1)}, \pi^{(k'+1)})$. $\qquad\square$

### A.3 PROOF FOR PROPOSITION 4.2

*Proof.* We start our proof by expressing (P2$'$) in a compact form as:

$$x^{(\varepsilon)} = \underset{\substack{x\geqslant 0 \\ Ax=b}}{\arg\min}\, \bar{c}^{\mathsf{T}}x + \tfrac{1}{\varepsilon}x^{\mathsf{T}}(\log x - \mathbf{1}_{(m+1)n})$$

where $x$ denotes the vector formed by stacking $\rho$ and the row vectors of $\pi$, $\bar{c}$ denotes the vector formed by stacking $n$-dimensional zero vector and the row vectors of $c$, and $A$ and $b$ are such that $Ax = b$ imposes transport constraints as:

$$A = \begin{bmatrix} I_{n\times n} & \overbrace{I_{n\times n} \quad \cdots \quad I_{n\times n}}^{m} \\ \mathbf{0}_n^{\mathsf{T}} & \mathbf{1}_{m\,n}^{\mathsf{T}} \end{bmatrix}, \quad b = [1/n\mathbf{1}_n^{\mathsf{T}} \quad \mu]^{\mathsf{T}}$$

From *Lagrangian dual*, we have:

$$x^{(\varepsilon)} = \exp(\text{-}\varepsilon(\bar{c}+A^{\mathsf{T}}\lambda^*))$$

where $\lambda^*$ is the optimal dual Lagrangian satisfying:

$$A\exp(\text{-}\varepsilon(\bar{c}+A^{\mathsf{T}}\lambda^*)) = b$$

Defining $[\frac{\partial x}{\partial c}]_{ij} \coloneqq \frac{\partial x_j}{\partial c_i}$, we have;

$$\frac{\partial x^{(\varepsilon)}}{\partial c} = -\varepsilon\bar{I}(I + \frac{\partial\lambda^*}{\partial c}A)Diag(x^{(\varepsilon)})$$

where $\bar{I} \coloneqq [\mathbf{0}_{(mn)\times n} \quad I_{(mn)\times((m+1)n)}]$. Similarly, for the dual variable, we have:

$$-\varepsilon(I + \frac{\partial\lambda^*}{\partial\bar{c}}A)Diag(x^{(\varepsilon)})A^{\mathsf{T}} = 0 \Rightarrow \frac{\partial\lambda^*}{\partial\bar{c}} = -Diag(x^{(\varepsilon)})A^{\mathsf{T}}(ADiag(x^{(\varepsilon)})A^{\mathsf{T}})^{\text{-}1}.$$

Putting back the expression for $\frac{\partial \lambda^*}{\partial \bar{c}}$ in $\frac{\partial x^{(\varepsilon)}}{\partial c}$, we obtain:

$$\frac{\partial x^{(\varepsilon)}}{\partial c} = -\varepsilon \bar{I}\big(Diag(x^{(\varepsilon)}) - Diag(x^{(\varepsilon)})A^\intercal(ADiag(x^{(\varepsilon)})A^\intercal)^{-1}ADiag(x^{(\varepsilon)})\big),$$

which includes $(m+1)$ by $n$ matrix inversion, $H := ADiag(x^{(\varepsilon)})A^\intercal$. We now show that $H^{-1}$ can be obtained without explicit matrix inversion.

$H$ can be expressed as:

$$H = \begin{bmatrix} 1/n I & 1/n - \rho \\ 1/n - \rho^\intercal & \mu \end{bmatrix}$$

$H$ is Hermitian and positive definite. Using block matrix inversion formula for such matrices (Corrolary 4.1 of (Lu & Shiou, 2002)), we obtain the inverse as;

$$H^{-1} = \begin{bmatrix} nI + k^{-1}\hat{\rho}\hat{\rho}^\intercal & -k^{-1}\hat{\rho} \\ -k^{-1}\hat{\rho}^\intercal & k^{-1} \end{bmatrix}$$

where $k = 1 - \mu - n\rho^{(\varepsilon)\intercal}\rho^{(\varepsilon)}$ and $\hat{\rho} = 1 - n\rho^{(\varepsilon)}$.

Given $\frac{\partial \mathcal{L}}{\partial x^{(\varepsilon)}}$, *i.e.*, $\big(\frac{\partial \mathcal{L}}{\partial \rho^{(\varepsilon)}}, \frac{\partial \mathcal{L}}{\partial \pi^{(\varepsilon)}}\big)$, the rest of the proof to obtain $\frac{\partial \mathcal{L}}{\partial c}$ follows from right multiplying the Jacobian, *i.e.*, $\frac{\partial \mathcal{L}}{\partial c} = \frac{\partial x^{(\varepsilon)}}{\partial c}\frac{\partial \mathcal{L}}{\partial x^{(\varepsilon)}}$ and rearranging the terms. $\qquad\square$

# B  OPTIMAL TRANSPORT BASED OPERATORS

In this section, we briefly discuss optimal transport based aggregation and selection operators. We provide their formulations to show how our formulation differs from them.

## B.1  FEATURE AGGREGATION

Given a cost map $c_{ij} = \|\omega_i - f_j\|_2$ which is an $m$ (number of prototypes) by $n$ (number of features) matrix representing the closeness of prototypes $\omega_i$ and features $f_j$, Mialon et al. (2021) consider the following optimal transport problem:

$$\pi^* = \arg\min_{\substack{\pi \geqslant 0 \\ \Sigma_i \pi_{ij} = 1/n \\ \Sigma_j \pi_{ij} = 1/m}} \sum_{ij} c_{ij}\pi_{ij} \tag{P4}$$

and defines their aggregated feature as:

$$g = \sqrt{m}[f_1 \mid f_2 \mid \cdots \mid f_n]\pi^\intercal . \tag{B.1}$$

Namely, $g$ is an ensemble of $m$ vectors each of which is the weighted aggregation of $\{f_i\}_{i \in [n]}$ with the weights proportional to the assignment weights to the corresponding prototype. In this ensemble representation, $g$, there is no feature selection mechanism and thus, all features are somehow included in the image representation.

If we further sum these $m$ vectors of $g$ to obtain a single global representation, we end up with global average pooling: $g^\intercal \mathbf{1}_m = \sqrt{m}[f_1 \mid f_2 \mid \cdots \mid f_n]\pi^\intercal \mathbf{1}_m = \sqrt{m}/n[f_1 \mid f_2 \mid \cdots \mid f_n]\mathbf{1}_n = \sqrt{m}/n\Sigma_i f_i$.

Briefly, Mialon et al. (2021) map a set of features to another set of features without discarding any and do not provide a natural way to aggregate the class-discriminative subset of the features. Such a representation is useful for structural matching. On the contrary, our formulation effectively enables learning to select discriminative features and maps a set of features to a single feature that is distilled from nuisance information.

## B.2  TOP-$k$ SELECTION

Given $n$-many scalars as $x = [x_i]_{i \in [n]}$ and $m$-many scalars as $y = [y_i]_{i \in [m]}$ with $y$ is in an increasing family, *i.e.*, $y_1 < y_2 < \ldots$, Xie et al. (2020) consider the following optimal transport problem:

$$\pi^* = \arg\min_{\substack{\pi \geqslant 0 \\ \Sigma_i \pi_{ij} = q_j \\ \Sigma_j \pi_{ij} = p_i}} \sum_{ij} c_{ij}\pi_{ij} \tag{P5}$$

where $c_{ij} = |y_i - x_j|$ and $p$ is $m$-dimensional probability simplex, *i.e.*, $p \in \{p \in \mathbb{R}^m_{\leq 0} \mid \Sigma_i p_i = 1\}$. Then, top-$k$ selection is formulated with the setting $q = 1/n\mathbf{1}_n$, $y = [0, 1]$ and $p = [\frac{k}{n} \; \frac{n-k}{n}]^\mathsf{T}$. Similarly, sorted top-$k$ selection is formulated with the setting $y = [k]$ and $p = [\frac{1}{n} \cdots \frac{1}{n} \; \frac{n-k}{n}]^\mathsf{T}$. Solving the same problem in (P5), Cuturi et al. (2019) formulate top-$k$ ranking by letting $q$ and $p$ be any probability simplex of the proper dimension and $y$ be in an increasing family.

Such top-$k$ formulations are suitable for selecting/ranking scalars. In our problem, the aim is to select a subset of features that are closest to the prototypes which are representatives for the discriminative information. Namely, we have a problem of subset selection from set-to-set distances. If we had our problem in the form of set-to-vector, then we would be able to formulate the problem using (P5). However, there is no natural extension of the methods in (Xie et al., 2020; Cuturi et al., 2019) to our problem. Therefore, we rigorously develop an optimal transport based formulation to express a discriminative subset selection operation analytically in a differentiable form.

With that being said, our formulation in (P1) differs from the typical optimal transport problem exploited in (P5). In optimal transport, one matches two distributions and transports all the mass from one to the other. Differently, we transport $\mu$ portion of the uniformly distributed masses to the prototypes that have no restriction on their mass distribution. In our formulation, we have a portion constraint instead of a target distribution constraint, and we use an additional decision variable, $\rho$, accounting for residual masses. If we do not have $\rho$ and set $\mu = 1$, then the problem becomes a specific case of an optimal transport barycenter problem with 1 distribution.

Our problem can be expressed in a compact form by absorbing $\rho$ into $\pi$ with zero costs associated in the formulation, which is indeed what we do in the proof of Proposition 4.2 (Appendix A.3). We choose to explicitly define $\rho$ in the problem (P1) to show its role and avoid convoluted notation. We believe its residual mass role is more understandable this way. The benefits of our formulation include that we can perform feature selection operation with matrix inversion free Jacobian and we can change the role of the prototypes as background representatives simply by using $\rho$ to weight the features instead of $1/n - \rho$ in Eq. (4.1). Our specific formulation further allows us to tailor a zero-shot regularization loss built on the learned prototypes within our pooling layer.

## C  IMPLEMENTATIONS WITH PSEUDO CODES

---

**Algorithm 1** Deep Metric Learning Loss with GSP and ZSR

---

**input:** $(X, Y) = (\{x_i\}, \{y_i\})_{i \in b}$ // a batch of image-label pairs
  $F \leftarrow \text{Backbone}(X)$                    // a CNN backbone such as *BN-Inception, ResNet*
  $(X_p, Z) \leftarrow \{\text{GSP}(f)\}_{f \in F}$        // get pooled features and attribute predictions, see Algorithm 2
  $\mathcal{L}_{ZSR} \leftarrow \text{ZSR}(Z, Y)$        // compute ZSR loss, see Algorithm 4
  $\mathcal{L}_{DML} \leftarrow \text{LossDML}(X_p, Y)$   // a DML loss such as *contrastive, triplet, XBM, LIBC, ...*
  $\mathcal{L} \leftarrow (1-\lambda)\mathcal{L}_{DML} + \lambda\mathcal{L}_{ZSR}$   // we set $\lambda$=0.1
**return** $\mathcal{L}$

---

**Algorithm 2** GSP($f$)

---

**trainable parameters:** $\omega = \{\omega_i\}_{i \in [m]}$     // $m$-many prototypes

---

**input:** $f = \{f_i\}_{i \in [n]}$                    // feature map, $n = w\,h$ (*i.e.*, width×height)
  $\bar{\omega}_i \leftarrow \omega_i / \max\{1, \|\omega_i\|_2\} \; \forall i \in [m], \; \bar{f}_j \leftarrow f_j / \max\{1, \|f_j\|_2\} \; \forall j \in [n]$
  $c_{ij} \leftarrow \|\bar{\omega}_i - \bar{f}_j\|_2$          // cost map, $c = \{c_{ij}\}_{(i,j) \in [m] \times [n]}$
  $\rho, \pi \leftarrow \text{WeightTransport}(c)$   // see Algorithm 3
  $f \leftarrow \frac{1 - n\rho}{\mu} \odot f$              // re-weight features, $\odot$: element-wise multiplication
  $x_p \leftarrow \big(\frac{1}{n} \sum\limits_{i \in [n]} f_i^p\big)^{1/p}$        // pooled feature, GSP for $p$=1, GeMean+GSP for $p$>1
  $z_i \leftarrow \frac{1}{\mu} \sum\limits_{j \in [n]} \pi_{ij} \; \forall i \in [m]$        // *pseudo-attribute* predictions, $z = \{z_i\}_{i \in [m]}$
**return** $x_p, z$

---

**Algorithm 3** WeightTransport($c$)

---

**hyperparameters:** $\mu$ : transport ratio, $\varepsilon$ : entropy regularization weight, $k$ : number of iterations

---

**forward:** gets cost map, $c$, returns residual weights, $\rho$, and transport plan $\pi$

---

**input:** $c = \{c_{ij}\}_{(i,j) \in [m] \times [n]}$        // cost map of $m$-many prototypes and $n$-many features
  $\kappa \leftarrow \exp(\text{-}\varepsilon c), \; t \leftarrow 1$        // exp is element-wise
  **repeat** $k$ times
    $\rho \leftarrow {}^1\!/n(1 + t\,\kappa^\intercal \mathbf{1}_m)^{-1}$        // $A^\intercal \mathbf{1}_m$: sum $A$ along rows, $(\cdot)^{-1}$ is element-wise
    $t \leftarrow \mu(\mathbf{1}_m^\intercal \kappa\,\rho)^{-1}$
**return** $\rho, \; t\,\kappa\,Diag(\rho)$        // $\pi \leftarrow t\,\kappa\,Diag(\rho)$

---

**backward:** gets the solution $(\rho, \pi)$ and the gradients $(\frac{\partial \mathcal{L}}{\partial \rho}, \frac{\partial \mathcal{L}}{\partial \pi})$, returns $\frac{\partial \mathcal{L}}{\partial c}$

---

**input:** $\rho, \; \pi, \; \frac{\partial \mathcal{L}}{\partial \rho}, \; \frac{\partial \mathcal{L}}{\partial \pi}$        // results of forward pass and the loss gradient w.r.t. them
  $q \leftarrow \rho \odot \frac{\partial \mathcal{L}}{\partial \rho} + (\pi \odot \frac{\partial \mathcal{L}}{\partial \pi})^\intercal \mathbf{1}_m$   // $A^\intercal \mathbf{1}_m$: sum $A$ along rows, $\odot$: element-wise multiplication
  $\eta \leftarrow (\rho \odot \frac{\partial \mathcal{L}}{\partial \rho})^\intercal \mathbf{1}_n - n\,q^\intercal \rho$
  $\frac{\partial \mathcal{L}}{\partial c} \leftarrow \text{-}\varepsilon\Big(\pi \odot \frac{\partial \mathcal{L}}{\partial \pi} - n\pi Diag\big(q - \frac{\eta}{1 - \mu - n\rho^\intercal \rho}\big)\rho\Big)$
**return** $\frac{\partial \mathcal{L}}{\partial c}$

---

---

**Algorithm 4** ZSR($Z, Y$)

---

**trainable parameters:** $\Upsilon = \{v_i\}_{i \in [\#\text{classes}]}$      // a semantic embedding vector for each class label

---

**input:** $Z = \{z_i\}_{i \in b}$, $Y = \{y_i\}_{i \in b}$   // a batch, $b$, of attribute prediction vectors, $z_i$, and their labels, $y_i$
     $(b_1, b_2) \leftarrow$ split $b$ into two class-disjoint halves s.t. $\{y_i\}_{i \in b_1} \cap \{y_i\}_{i \in b_2} = \emptyset$
     $\Upsilon_k \leftarrow [v_{y_i}]_{i \in b_k}$ for $k{=}1, 2$      // label embedding matrix for batch-$k$, *i.e.*, prediction targets
     $Z_k \leftarrow [z_i]_{i \in b_k}$ for $k{=}1, 2$      // attribute prediction matrix for batch-$k$, *i.e.*, prediction inputs
     $A_k \leftarrow \Upsilon_k (Z_k^{\mathsf{T}} Z_k + \epsilon I)^{-1} Z_k^{\mathsf{T}}$ for $k{=}1, 2$    // fit label embedding predictor for batch-$k$, $\epsilon{=}0.05$
     $\hat{\Upsilon}_1 \leftarrow A_2 Z_1$, $\hat{\Upsilon}_2 \leftarrow A_1 Z_2$      // use predictor for $b_k$ to predict the label embeddings of $b_{k'}$
     $\hat{\Upsilon} \leftarrow [\hat{\Upsilon}_1 \mid \hat{\Upsilon}_2]$      // concatenate predictions
     $S \leftarrow \text{SoftMax}(\hat{\Upsilon}^{\mathsf{T}} \Upsilon)$      // similarity scores between predictions and label embeddings
     $\mathcal{L}_{ZSR} \leftarrow \text{CrossEntropy}(S, Y)$
**return** $\mathcal{L}_{ZSR}$

---

