# OpenReview forum: "Generalized Sum Pooling for Metric Learning"
_ICLR.cc/2023/Conference — Submitted to ICLR 2023_

### Official Review · Reviewer_MQSC · 2022-10-16

**Confidence:** 4
**Correctness:** 3
**Technical Novelty And Significance:** 3
**Empirical Novelty And Significance:** 3
**Recommendation:** 6

**Clarity, Quality, Novelty And Reproducibility:**

The proposed Generalized Sum Pooling (GSP) operation and zero-shot regularization are novel, and a feasible solution is given to the optimization problem.

Experiments support the effectiveness of the proposed GSP and zero-shot regularization methods.

Code is provided for reproduction.

Overall, the quality of this is good, except for some concerns listed in the above Weaknesses.

**Strength And Weaknesses:**

## Strength
- The proposed OT optimization problem in (P1) is novel and reasonable as an alternative to GAP. The introduction of $\rho$ and $\mu$ are the keys to model nuisance features and dynamic re-weighting.
- Both forward and backward solutions are given for an entropy-regularized version of (P1), i.e., (P2). Moreover, the closed-form and inversion-free formulation of the partial derivatives enables differentiable and efficient backward computation.
- A zero-shot regularization auxiliary task is proposed to improve the transferring ability of metric learning. A simple formulation with a closed-form solution is devised, and a meta-learning procedure is proposed to enable training without ground truth label embeddings.
- The toy experiments corroborate the design intuitions of GSP and zero-shot regularization, providing useful insights. The comparison experiments are conducted on various datasets, with 13 pooling alternatives and with various backbone/methods, verifying the general effectiveness.

## Weakness
- One major concern is the extra parameters and computation complexities induced by the proposed method. For example, extra parameters involve $\{ \omega, \Upsilon \}$ and extra computations involve Proposition 4.1 ($k=100$ forward iteration) and Proposition 4.2 (backward computation).
- For the deriving of (4.2), important details are missing, e.g., $\frac{\partial \mathcal{L}}{\partial \rho^{(\epsilon)}}$. In $\frac{\partial \mathcal{L}}{\partial c}$, which kind of $\mathcal{L}$ is used? In Proposition 4.2, the first $\mathcal{1}_n$ should be $\mathcal{1}_m$, and $\gamma$ is not defined. All these make 4.2 incomplete.
- I can infer the procedure of meta-learning in zero-shot regularization from Figure 2. However, Section 4.3 is a bit confusing, especially (4.4). It will be much better if an algorithm box is provided.
- The author mentioned several times the similarity with the top-$k$ selection process. However, no discussion or formulation analysis is provided to show the relationship explicitly.
- In Sec 4.2, "this function is not smooth" why? Is it because of the max operation in normalization?
- There are several typos, especially in the appendix regarding Preposition 4.1.



**Summary Of The Paper:**

This paper proposes a learnable and differentiable pooling operation to replace the global average pooling (GAP) for metric learning.

Specifically, the pooling operation is formulated as the solution to an optimal transport problem with additional residual $\rho$ and mass ration $\mu$ parameters. $\rho$ represents the per-element residual, and $\mu$ stands for the total assignment mass.
The authors derived an iterative solution for forward computation and an inversion-free computation of gradient for backward computation. The closed-form backward gradient formulation is efficient for network training.

Moreover, to improve the transferring property, cross-batch zero-shot regularization is proposed by predicting class embedding vectors from the prototype assignment vectors ($\pi$). The sub-problem (P3) has a closed-form solution. Since real class embedding vectors are not given, a meta-learning scheme utilized a zero-shot prediction loss to tackle this.

In the experiments, toy experiments are designed to investigate the mechanism of the proposed GSP. In metric learning experiments, various pooling alternatives are compared, and GSP achieves the best performance. In addition, GSP is evaluated under different architectures and methods, verifying its effectiveness. Regarding the absolute improvements, the general boost is around or below 1% in most cases, but the improvements are consistent across all settings.




**Summary Of The Review:**

The overall quality and novelty of this paper are good, except for several concerns listed above. The experiments can verify the effectiveness of the proposed method.
I would like to see this paper accepted if the concerns and issues are solved by the authors.

---

> ### Author Response · Authors · 2022-11-09
> **Our response to R-MQSC**
>
> We thank the reviewer (**R-MQSC**) for their constructive thorough detailed review and acknowledgment to our contributions. We appreciate the encouraging comments. We will use the comments and suggestions to improve the quality and presentation of the paper. We address the specific concerns as follows:
>
> ### [...One major concern is the extra parameters and computation complexities…]
>
> **Parameters:** Extra parameters are only as many as two $1\times 1$ convolution layers have. Specifically, our **GSP** layer **only introduces an $m$-kernel $1\times 1$ convolution layer** (*i.e.,* $m$-many trainable prototypes) and similarly, zero-shot regularization (**ZSR**) loss **introduce $c$-kernel $1\times 1$ convolution layer** (*i.e.,* $c$-class classification layer). For example, for the DML problem using BN-Inception backbone in CUB, **this would increase the number of parameters from $\approx$ 10M to 10.03M**, so it is **negligible**.
>
> **Computation:** Our **additional computational complexity cost is minor**, specifically: $\approx$ %7-8.5 increase (Fig.-7: time per batch up to 120 ms - 302 ms from 112 ms - 278 ms).
>
> As we summarize in Algorithm-2 and -3, the repeated operations of GSP involve summing a matrix along its rows, a dot product and element-wise reciprocal of a vector, which **can be computed quite efficiently**. For the **backward computation**, we only have **a few lines of elementary operations**.
>
> For ZSR, we only need to invert matrices of $(batch size)\times (batch size) $ which is typically small (*e.g.,* 32$\times$32). For the settings with large batch size, we can use Woodbury identity to obtain an equivalent expression involving matrices with size $m \times m$, *i.e.*,  $(num. prototypes) \times (num. prototypes)$.
>
> ### Similarity with Top-$k$ Selection
>
> We **extended the supplementary material and added the related discussions to Section-B**. Briefly, **our formulation** in (P1) **selects** a subset of the **features** that are closest to some prototypes while **top-$k$ operators select** a subset of **scalars**. In other words, our formulation solves the problem of subset selection from set-to-set distances while top-$k$ operators can solve subset selection from set-to-vector distances.
>
> ### About Proposition-4.2
>
> Proposition-4.2 is true for any differentiable loss function. We fixed the statement accordingly with the general form.
>
> **Typos:** There existed a few typos confusing the expression. $\gamma$ should have been $\varepsilon$ and the first $\mathbf{1}_n$ should have been $\mathbf{1}_m$ as the reviewer points out.
>
> ### Other Comments
>
> **Algorithm boxes.** We **extended the supplementary material with the algorithmic boxes** for the implementation of our framework.
>
> **Function is not smooth.** $\rho$ as a function of $\omega$ is not smooth since $\rho$ is a function of $c$ through the linear program (P1) which is not smooth. On the other hand, $c$ as function of $\omega$ is smooth since so is the normalization operation.
>
> **Typos.** We have corrected the typos in the revised version of the paper.

---

### Official Review · Reviewer_twxc · 2022-10-21

**Confidence:** 3
**Correctness:** 3
**Technical Novelty And Significance:** 2
**Empirical Novelty And Significance:** 2
**Recommendation:** 5

**Clarity, Quality, Novelty And Reproducibility:**

Clarity: This paper is mostly well-written.


Quality: Experiments do not successfully show the effectiveness of GSP. The standalone performance of GSP is not better than several existing pooling methods.


Novelty: Algorithmic novelty is lacking. The idea of learning pooling layer with optimal transportation has already been proposed. The problems (P1, P2) are equivalent to the basic optimal transportation formulations, so the solution is also not novel.


Reproducibility: Code is provided.


**Strength And Weaknesses:**

------------------- Strengths -------------------

1. Introducing background prototype in optimal transportation problem is interesting.


2. Thorough experiments are reported (in Supplementary Material).


3. The paper is mostly well-written.


4. Code is provided.

------------------ Weaknesses ------------------

1. Novelty

Algorithmic novelty may not be sufficient. As discussed in the paper, the idea of using optimal transportation formulation for feature pooling was proposed in Mialon et al. (2021), and the proposed problem (P1) is equivalent to a standard optimal transportation problem (by considering rho as a new pi_i, we can immediately get to the standard form). Given these, using entropy regularization, which is an efficient approach to solving the optimal transportation problem, is straightforward. (P2) is also equivalent to the standard form of the optimal transportation problem. The same is applied to the algorithm.


2. Motivation

While this paper focuses on metric learning, the application of pooling in general should not be limited to this. So far, I could not find a good reason for this limitation. I would expect an explanation if any.


3. Evaluation of Rho

The point of GSP is to use rho, in other words, changing the range of i in (P1) 1<=i<=m to 1<=i<=m+1. So a before and after comparison between these two cases would be necessary.


4. Overall Performance

The results of comparisons with various pooling methods reported in Supplementary Material (e.g., Table 2) show that the stand-alone performance of GSP is outperformed by several existing methods.


5. Hyperparameter Setting

GSP, together with zero-shot regularization, has many hyperparameters to be tuned (m, mu, epsilon, k, lambda). Since these were likely tuned for each dataset and task, it would not have been very difficult to achieve equal or better accuracy than other methods.


**Summary Of The Paper:**

An extension of the global average pooling (GAP), called generalized sum pooling (GSP), is presented in this paper: while GAP is a convex combination with "flat weights," GSP instead learns the weights by solving an optimal transportation problem. GSP allows for the selection of features to be pooled by introducing a "background prototype" (rho) into the formulation. Experiments compared GSP with several existing metric learning and pooling methods. The results showed that GSP gives small improvements in several cases.

**Summary Of The Review:**

The idea of using background prototypes is interesting and thorough experiments have been reported.
The major problems with this paper are the lack of novelty in the algorithm and the limited effectiveness of the proposed method. Also, the proposed method has many hyperparameters, which may be a drawback for its use in practical scenarios. Given that both novelty and effectiveness are important, I am leaning toward rejection for now.

---

> ### Author Response · Authors · 2022-11-09
> **Our response to R-twxc (3/3)**
>
> ## Comments on Effectiveness
>
> ### [...Experiments do not successfully show the effectiveness of GSP…]
>
> As **R-MQSC** also **points out**, we conduct comparison experiments on various datasets, with various pooling alternatives and with various backbone/methods to **verify the general effectiveness**.
>
> The results show that **our improvements are consistent across all settings**. GSP consistently obtains the best performance except for a few cases in metric learning benchmarks where we perform on par with GeMean (Radenovic et al., 2018).
>
> With that being said, in *Cifar Collage* dataset where **ignoring nuisance information is important**, the performances show **substantial superiority of our method's discriminative feature pooling mechanism** to GeMean. As we discuss in the paper, GSP can be applied to GeMean to further boost the performance, which is also empirically validated.
>
> ### [...method has many hyperparameters, which may be a drawback for its use in practical scenarios…]
>
> According to our analysis which is provided in supplementary material Section-1.6, the **performance** of our method is **not sensitive to the hyperparameter setting**. To summarize:
>
> + $m$: We show that any $m\geq 64$ works well in practice (Fig.-8) and tuning it only brings marginal differences. $m$ can be considered as deciding the number of kernels in a convolution layer.
>
> + $\mu$: Entropy regularization reduces susceptibility to $\mu$ parameter. As long as it is not close to 1.0, we obtain similar performances as we validate experimentally (Fig.-6).
>
> + $k$: We set $k$ to a large value $\approx 50-100$ since its increase does not increase the computation substantially (Fig.-7) thanks to closed form gradient expression.
>
> + $\varepsilon:$ According to our analysis, we suggest $\varepsilon=5.0$ for non-proxy losses and $\varepsilon=0.5$ for proxy-based losses.
>
> + $\lambda$: We always fix it to 0.1.
>
> As we also disclose in the paper, **we do not fine tune hyperparameters to each scenario**. We **fix the parameters across all the settings** after our analysis. Having **consistent performance** improvements **across all the datasets and methods**, we believe that one can directly use our **suggested settings** in their applications or one can **easily adapt the settings** to their application by **building on our analysis** in the supplementary material $\S$-1.6.
>
>
> ## Other Comments
>
> ### [...this paper focuses on metric learning…I could not find a good reason for this limitation…]
>
> We would like to point out that almost all SOTA methods in metric learning utilize a form of GAP. In order to improve metric learning performance, improving GAP is a rather sensible approach. Moreover, metric learning is an important problem with a large audience within ICLR; hence, it is clearly not a limitation.
>
>
> ### Evaluation of $\rho$
>
> **Our empirical study involves the related experiments** that the reviewer asks. Specifically, we provide the evaluation of zero-residual (*i.e.* $\mu=1$) case with zero-shot prediction loss on metric learning benchmark datasets in Table-1 and evaluation on *Cifar Collage* dataset in Fig.-3(b).
>
> We should note that if we do not use $\rho$ in (P1), *i.e.* $i\in[m]$, then we must have $\mu=1$ or the problem will be infeasible otherwise. Once we set $\mu = 1$, we will have no residual weights on the features and thus, GSP would be equivalent to global average pooling. Moreover, we already compare with GAP.

---

> ### Author Response · Authors · 2022-11-09
> **Our response to R-twxc (2/3)**
>
> ## Comments on Novelty (2/2)
> ### [...the proposed problem (P1) is equivalent to a standard optimal transportation problem…]
>
> In optimal transport, one matches two distributions and transports all the mass from one to the other. Differently, we transport $\mu$ portion of the uniformly distributed masses to the prototypes that have **no restriction** on their mass **distribution**.
>
> In our formulation, we have **a portion constraint instead of a target distribution constraint**, and we use an additional decision variable, $\rho$, accounting for residual masses. If we do not have $\rho$ and set $\mu = 1$, then the problem becomes **a specific case of an optimal transport barycenter problem with 1 distribution** which has not been applied to pooling before. Hence, absorbing $\rho$ into $\pi$ **does not give a standard optimal transport formulation**.
>
> Moreover, **optimal transport** based top-$k$ formulations are **suitable for selecting/ranking scalars**. In our problem, **the aim is to select a subset of vectors** that are closest to the prototypes which are representatives for the discriminative information. Hence, we have a problem of **subset selection from set-to-set distances**.
>
> If we had our problem in the form of set-to-vector, then we would be able to formulate the problem using optimal transport. However, there is **no natural extension of the optimal transport based methods** that we cover in the related work to our problem. Therefore, we rigorously develop an optimal transport based formulation to express a discriminative subset selection operation analytically in a differentiable form.
>
> The **benefits of our formulation** include that we can perform **feature selection** operation with **matrix inversion free Jacobian** and we can change the role of the prototypes as background representatives simply by using $\rho$ to weight the features instead of $\tfrac{1}{n} - \rho$ in Eqn. (4.1). Our specific **formulation further allows** us to tailor **a zero-shot regularization loss** built on the learned prototypes within our pooling layer.

---

> ### Author Response · Authors · 2022-11-09
> **Our response to R-twxc (1/3)**
>
> We thank the reviewer (**R-twxc**) for their time and valuable feedback. We appreciate the encouraging comments and acknowledgement to our contributions. We will use the comments and suggestions  to improve the quality and presentation of the paper. We address the specific concerns as follows:
>
> ## Comments on Novelty (1/2)
>
> ### Our method is substantially different from the one proposed in (Mialon et al., 2021)
>
> | method                       | Mialon et al. (2021)                                                                                                                                             | Ours                                                                                                                                                                    |
> |------------------------------|------------------------------------------------------------------------------------------------------------------------------------------------------------------|-------------------------------------------------------------------------------------------------------------------------------------------------------------------------|
> | optimization problem         | $\underset{\substack{\pi\geq 0 \newline \Sigma_i \pi_{ij} = \tfrac{1}{n} \newline \Sigma_j \pi_{ij} = \tfrac{1}{m}}}{\mathrm{argmin}} \Sigma_{ij}c_{ij}\pi_{ij}$ | $\underset{\substack{\rho, \pi\geq 0 \newline \rho_j+\Sigma_i \pi_{ij} = \tfrac{1}{n} \newline \Sigma_{ij} \pi_{ij} = \mu}}{\mathrm{argmin}} \Sigma_{ij}c_{ij}\pi_{ij}$ |
> | image representation         | $\sqrt{m}[f_1 \mid f_2 \mid \cdots\mid f_n]\pi^{\intercal}\quad$                                                                                                  | $\Sigma_i \tfrac{1 - n\rho_i}{n\mu}f_i$                                                                                                                                 |
> | dimension                    | $m\times d$                                                                                                                                                      | $d$                                                                                                                                                                     |
> | feature selection            | ❌                                                                                                                                                       | ✔️                                                                                                                                                            |
> | gradient computation         | auto-diff                                                                                                                                                        | closed form expression                                                                                                                                                  |
> | matrix-inverse-free gradient | ❌                                                                                                                                            | ✔️                                                                                                                                                            |
>
> As we discuss in the related work as well as in the revised supplementary material $\S$-B, **Mialon et al. (2021)** solve an optimal transport problem and **form an ensemble of aggregated features**. In such a representation, if the dimension of the aggregated features is $d$ (*i.e.,* $f_i\in\mathbb{R}^d$), then the dimension of the image representation is $m\times d$. Their aim is to form an image representation so that $\ell_2$ distances between such representations approximately reflect optimal transport distance in between. There is **no feature selection mechanism** and thus, all features are somehow included in the image representation.
>
> Briefly, Mialon et al. (2021) **map a set of features to another set of features without discarding any** and do **not provide a natural way to aggregate the class-discriminative subset** of the features. On the contrary, **our formulation effectively enables learning to select discriminative features** and **maps a set of features to a single feature that is distilled from nuisance information**.
>
> We empirically show in Table-4 that **we outperform** Mialon et al. (2021)'s pooling method **by large margin**. In particular, *Cifar Collage* experiments clearly show the **superiority of our method's feature selection mechanism**. We also show in CUB dataset that **Mialon et al. (2021)'s method demands larger features** sizes to perform on par. Moreover, **Mialon et al. (2021) rely on automatic differentiation** during the backpropagation through their pooling layer while **we can use matrix-inversion-free closed form expression**. We empirically show in Fig.-7 that using **auto-diff increases computation by %34-54.5**.

---

### Official Review · Reviewer_ms1e · 2022-10-25

**Confidence:** 3
**Correctness:** 3
**Technical Novelty And Significance:** 3
**Empirical Novelty And Significance:** 3
**Recommendation:** 5

**Clarity, Quality, Novelty And Reproducibility:**

Code is not available at this stage, - a password is required.

I'm not an expert on this area, hence I would it seems to be novel.



**Details Of Ethics Concerns:**

N.A.

**Strength And Weaknesses:**

I'm not an expert in this area, please correct me if I get it wrong.

Strength:
1. The proposed method seems to be sound, but I'm not fully checking it. The experimental results are outstanding
2. The motivation seems to be reasonable.

Weakness:
1. While the motivation is reasonable, it still confuses me why we should redefine the learnable generalization of GAP as an optimization problem. What is the optimization objective?
2. It seems like there are three categories of related works that are mentioned in this paper,  when selecting the baselines in your experiments, do you cover all of those three categories?
3. Given the complicated structure of the GSP (i.e., Fig 2), how to can apply it to current existing metric learning or CV-related works?
4. It is not clear to me, why we should bring the OT into this paper?

**Summary Of The Paper:**

This paper proposes a new pooling technique named generalized sum pooling. It is claimed as a strict generalization of global average pooling. The experiments are conducted on four different datasets and achieve a SOTA performance.

**Summary Of The Review:**

Based on my above point, I think this paper is novel and sound, but I do not fully check it through.
While it still misses some key points which makes me confuse.

---

> ### Author Response · Authors · 2022-11-09
> **Our response to R-ms1e**
>
> We thank the reviewer (**R-twxc**) for their time and valuable feedback. We appreciate the encouraging comments and acknowledgement to our contributions. We will use the comments and suggestions to improve the quality and presentation of the paper. We address the specific concerns as follows:
>
> ### [...why we should redefine the learnable generalization of GAP as an optimization problem…]
>
> It is important to notice that **GAP in its original form** can be **expressed as a trivial optimization problem** with the objective "*select $n$ features among $n$-many features, $\{f_i\}_{i\in[n]}$, and sum them*". Formally, this equivalence corresponds to $f_{GAP} = [f_1 \mid f_2 \mid \cdots \mid f_n] \pi^{\ast \intercal} \mathbf{1}_m $  where
>
> $$
> \pi^\ast = \underset{\substack{\pi\geq 0 \newline \Sigma_i \pi_{ij}=\tfrac{1}{n}\newline \Sigma_{ij}\pi_{ij}=1}}{\mathrm{argmin} \Sigma_{ij}0\pi_{ij}}  \quad\quad\quad (1)
> $$
>
> Hence, **generalizing the optimization objective** with "*select $n^\prime \leq n$ features among $n$-many features*" is **a sensible way to generalize GAP**. In other words, **we redefine** the learnable generalization of GAP **as an optimization problem** since **its original form can also be expressed as an optimization problem**. We formally **extend this optimization problem into a learnable form** as **one of our main contributions**. Moreover, since our solver is analytical, this re-definition does not bring any computational difficulty.
>
>
> ### [...What is the optimization objective?...]
>
> Consider we are given $m$-many feature prototypes which are representatives for the discriminative information. Our **objective is to select** the $\mu\approx \tfrac{n^\prime}{n}$ portion of the **features** that are closest to those prototypes. As we explain in Section-4.1, we formally express such an objective as:
> $$
> \rho^\ast, \pi^\ast = \underset{\substack{\rho, \pi\geq 0 \newline \rho_j + \Sigma_i \pi_{ij}=\tfrac{1}{n}\newline \Sigma_{ij}\pi_{ij}=\mu}}{\mathrm{argmin} \Sigma_{ij}c_{ij}\pi_{ij}}  \quad\quad\quad (2)
> $$
> where the **optimization objective is to transfer $\mu$ portion of the flat weights from the features to the prototypes with minimum cost**, where the cost, $c_{ij}$, is the distance between feature-$j$ and prototype-$i$. We then use residual weights $\rho$ to pool the features as we explain in the paper (Eqn. (4.1)).
>
> ### [...why we should bring the OT into this paper?...]
>
> We are **not artificially adding OT to the paper**. In fact, we theoretically show **GAP is closely relevant to OT**; hence, **its generalization** is also **closely related to OT**.  We naturally use tools from OT to solve this generalization.
>
>
> ### [...do you cover all of those three categories?...]
>
> Yes, we cover all three categories discussed in the related work in our experiments. Specifically:
>
> **DML.** We cover the fundamental loss terms that many deep metric learning (DML) methods employ in Table-2. Specifically, we include *contrastive, triplet, multi-similarity, proxy-nca, proxy-anchor* losses in our experiments. Moreover, we apply our method with SOTA *XBM* and *LIBC* DML approaches and provide the evaluation results in  Table-2 (XBM) and Table-3 (XBM and LIBC).
>
> **Prototype-based pooling.** We evaluate prototype-based VLAD and optimal transport pooling (OTP) in metric learning task and provide the results in Table-4.
>
> **Attention-based pooling.** We compare our method with 13 pooling alternatives in Table-4. Among those methods, we include attention-based CroW, Trainable-SMK, CBAM, DeLF, and GSoP.
>
> ### [...Given the complicated structure of the GSP (i.e., Fig 2), how to can apply it to current existing metric learning or CV-related works?...]
>
> The **implementation of our pooling layer** within a NN architecture is **a few lines of code composed of elementary operations**. We provide pseudo code implementations of our method in the revised version of the paper (in supplementary material Section-C). As implemented in Algorithm-1, **GSP simply can replace any global pooling layer**.

---

### Author Response · Authors · 2022-11-09
**Summary of Revisions**

We thank all the reviewers for their time and effort spent providing valuable feedback. Building on their comments as well as expanding on the discussion points in our responses, we have made the following changes to the manuscript:

+ **Further Discussions:** We devote a section in supplementary material (Section-B) to expand on the discussions of the optimal transport based aggregation and top-$k$ operators included in the related work (Section-2).

+ **Algorithm Boxes:** We provide algorithmic boxes in supplementary material Section-C to summarize the implementation of our method and to improve the presentation quality of the paper.

+ **Fixed Proposition-4.2:** We corrected a few typos in Proposition 4.2 as well as in Appendix (supplementary material Section-A) as we mentioned in our response to **R-MQSC**.

+ **Typos:** We corrected a typo regarding the performance of GSP with contrastive loss (C2+GSP) in CUB dataset (Table-2). We should note that the same experimental result (128D-MAP@R performance) is correctly recorded in Table-4 in the initial version of the paper.

---

### Decision · Program_Chairs · 2023-01-20

**Decision:**

Reject

**Justification For Why Not Higher Score:**

After reading this paper and the comments, I tend to reject it because of the following reasons.

(1) As some reviewers mentioned, OT-based global pooling has been proposed in some existing work [a, b, c]. The authors did not compare their method with these existing solutions. Although the authors highlighted the difference between their work and [a] in the rebuttal phase and claimed that their method corresponds to an unbalanced OT problem, this setting has been used in [b].

(2) One technical contribution of this method is deriving the gradient of OT in a closed form, which reduces the memory cost compared to the autodiff method. However, the closed-form gradient of OT has been derived in [d, e], which is not new for the community. The difference between the proposed derivation and [d, e] is trivial.

(3) As shown in the experimental results, the improvements caused by the proposed GSP method are incremental. At the same time, the computational cost introduced by GSP, in my opinion, is high because of solving the entropic OT problem. Additionally, I think more analytic experiments can be added. (1) It would be nice if the authors could visualize the optimal transport learned by the proposed method and provide some interpretability for the result. (2) The influence of the number of iterations "k" should be analyzed.

[a] Mialon, Grégoire, et al. "A trainable optimal transport embedding for feature aggregation and its relationship to attention." arXiv preprint arXiv:2006.12065 (2020).

[b] Cheng, M., & Xu, H. (2022). Revisiting Pooling through the Lens of Optimal Transport. arXiv preprint arXiv:2201.09191.

[c] Naderializadeh, Navid, et al. "Pooling by sliced-Wasserstein embedding." Advances in Neural Information Processing Systems 34 (2021): 3389-3400.

[d] Gould, Stephen, Richard Hartley, and Dylan Campbell. "Deep declarative networks." IEEE Transactions on Pattern Analysis and Machine Intelligence 44.8 (2021): 3988-4004.

[e] Xie, Yujia, et al. "A hypergradient approach to robust regression without correspondence." arXiv preprint arXiv:2012.00123 (2020).

**Justification For Why Not Lower Score:**

N/A

**Metareview: Summary, Strengths And Weaknesses:**

In this paper, a generalized pooling method is proposed for metric learning.
Essentially, the proposed method extends the expectation-maximization framework used by GAP, whose implementation is based on the well-known entropic optimal transport algorithm. To verify the feasibility of the proposed method, the authors tested their pooling method in metric learning tasks and compared it to other pooling strategies.

Strengths:
(1) The idea of the proposed method is reasonable, and its derivation is clear and easy to follow.

Weaknesses:
(1) The novelty of the proposed method is limited because many OT-based pooling/fusion methods have been proposed for recent two years (see the references [a-e] listed below). In the aspect of methodology and implementation, the differences between the proposed method and existing ones are not significant, in my opinion.
(2) The experimental results are not strong enough. The improvements achieved by the proposed method are incremental in most situations.

**Summary Of Ac-Reviewer Meeting:**

N/A